# On Vanishing Gradients, Over-Smoothing, and Over-Squashing in GNNs: Bridging Recurrent and Graph Learning

**Álvaro Arroyo**[1,*]  **Alessio Gravina**[2,*]  **Benjamin Gutteridge**[1]  **Federico Barbero**[1]
**Claudio Gallicchio**[2]  **Xiaowen Dong**[1]  **Michael Bronstein**[1,3]  **Pierre Vandergheynst**[4]

[1]University of Oxford  [2]University of Pisa  [3]AITHYRA  [4]EPFL

## Abstract

Graph Neural Networks (GNNs) are models that leverage the graph structure to transmit information between nodes, typically through the message-passing operation. While widely successful, this approach is well-known to suffer from representational collapse as the number of layers increases and insensitivity to the information contained at distant and poorly connected nodes. In this paper, we present a unified view of the appearance of these issues through the lens of *vanishing gradients*, using ideas from linear control theory for our analysis. We propose an interpretation of GNNs as recurrent models and empirically demonstrate that a simple state-space formulation of a GNN effectively alleviates these issues *at no extra trainable parameter cost*. Further, we show theoretically and empirically that (i) Traditional GNNs are by design prone to extreme gradient vanishing even after a few layers; (ii) Feature collapse is directly related to the mechanism causing vanishing gradients; (iii) Long-range modeling is most easily achieved by a combination of graph rewiring and vanishing gradient mitigation. We believe our work will help bridge the gap between the recurrent and graph learning literature and unlock the design of new effective models that benefit from both worlds.

## 1 Introduction

Graph Neural Networks (GNNs) [98, 45, 94, 76, 17, 29] have become a widely used architecture for processing information on graph domains. Most GNNs operate via *message passing*, where information is exchanged between neighboring nodes, giving rise to Message-Passing Neural Networks (MPNNs). Some of the most popular instances of this type of architecture include GCN [67], GAT [101], GIN [106], and GraphSAGE [56].

Despite its widespread use, this paradigm also suffers from some fundamental limitations. Most importantly, we highlight the issue of feature collapse (sometimes termed as *over-smoothing*) [79, 18], where feature representations become exponentially similar as the number of layers increases, and *over-squashing* [1, 100, 30], which describes the difficulty of propagating information across faraway nodes, as the exponential growth in a node's receptive field results in many messages being compressed into fixed-size vectors. Although these two issues have been studied extensively, and there exists evidence that they are trade-offs of each other [44], there is no unified theoretical framework that explains *why architectures that solve these problems work* and whether there exists a common *underlying cause* that governs these problems.

In this work, we analyze these issues arising from GNN depth through the lens of **vanishing gradients**. In particular, we ask several questions regarding the appearance and consequences of this

---

*Equal contribution. Correspondance to alvaro.arroyo@univ.ox.ac.uk

39th Conference on Neural Information Processing Systems (NeurIPS 2025).

phenomenon in GNNs: (i) How prone are GNNs to gradient vanishing? (ii) What is the effect of gradient vanishing on node feature evolution? (iii) Can preventing vanishing gradients effectively mitigate over-squashing and enable long-range modelling? (iv) Can methods used in the non-linear [61, 82, 9] and more recently, linear [52, 81] recurrent neural network (RNN) literature be effective at dealing with feature collapse and long-range modeling? We answer these questions by providing a novel perspective that bridges the gap between recurrent and graph learning.

**Contributions and outline.** In summary, the contributions of this work are the following:

- In Section 3, we explore the connection between GNNs and recurrent models and demonstrate how classical GNNs are susceptible to a phenomenon we term *extreme gradient vanishing*. We propose GNN-SSM, a GNN model that is written as a state-space model, allowing for better control of the spectrum of the Jacobian.

- In Section 4, we show how vanishing gradients contribute to node feature evolution, providing a more precise explanation for why GNNs struggle with depth and how node feature collapse emerges via spectral analysis of the layer-wise Jacobians. We show that GNN-SSMs are able to *exactly* control the rate of collapse of deep representations.

- In Section 5, we show how vanishing gradients are related to over-squashing. We argue that over-squashing should therefore be tackled by approaches that both perform graph rewiring *and* mitigate vanishing gradients.

Overall, we believe that our work provides a new and interesting perspective on well-known problems that occur in GNNs, from the point of view of sequence models. We believe this to be an important observation connecting two very wide – yet surprisingly disjoint – bodies of literature.

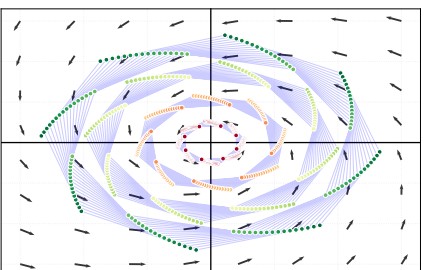

Figure 1: Latent evolution of 2-dimensional node features when passing through layers of a GNN-SSM with $\text{eig}(\Lambda) \approx 1$. Node states evolve in a norm-preserving manner, without collapsing or contracting. The blue lines indicate how each node feature evolves across layers, i.e., as more layers are added. Each circle corresponds to a node's 2D feature. Circles connected by a blue line represent the same node across successive layers. The color of each circle encodes the norm of the node feature, and the vector field indicates direction.

## 2 Background and Related Work

We start by providing the required background on graph and sequence models. We further discuss the existing literature on over-smoothing and over-squashing in GNNs and vanishing gradients in recurrent sequence models.

### 2.1 Message Passing Neural Networks

Let a graph $G$ be a tuple $(V, E)$ where $V$ is the set of nodes and $E$ is the set of edges. We denote edge from node $u \in V$ to node $v \in V$ with $(u, v) \in E$. The connectivity structure of the graph is encoded through an *adjacency matrix* defined as $\mathbf{A} \in \mathbb{R}^{n \times n}$ where $n$ is the number of nodes in the graph. We assume that $G$ is an undirected graph and that there is a set of feature vectors $\{\mathbf{h}_v\}_{v \in V} \in \mathbb{R}^d$, with each feature vector being associated with a node in the graph. Graph Neural Networks (GNNs) are functions $f_{\boldsymbol{\theta}} : (G, \{\mathbf{h}_v\}) \mapsto \mathbf{y}$, with parameters $\boldsymbol{\theta}$ trained via gradient descent and $\mathbf{y}$ being a node-level or graph level prediction label. These models typically take the form of Message Passing Neural Networks (MPNNs), which compute latent representation by composing $K$ layers of the following node-wise operation:

$$\mathbf{h}_u^{(k)} = \phi^{(k)}(\mathbf{h}_u^{(k-1)}, \psi^{(k)}(\{\mathbf{h}_v^{(k-1)} : (u, v) \in E\})), \tag{1}$$

for $k = \{1, \ldots, K\}$, where $\psi^{(k)}$ is a *permutation-invariant aggregation function* and $\phi^{(k)}$ *combines* the incoming messages from one's neighbors with the previous embedding of oneself to produce an updated representation. The most commonly used aggregation function takes the form

$$\psi^{(k)}(\{\mathbf{h}_v^{(k-1)} : (u, v) \in E\}) = \sum_u \tilde{\mathbf{A}}_{uv} \mathbf{h}_v^{(k-1)} \tag{2}$$

where $\tilde{\mathbf{A}} = \mathbf{D}^{-\frac{1}{2}}\mathbf{A}\mathbf{D}^{-\frac{1}{2}}$, and $\mathbf{D} \in \mathbb{R}^{n \times n}$ is a diagonal matrix where $\mathbf{D}_{ii} = \sum_j \mathbf{A}_{ij}$. One can also consider a matrix representation of the features $\mathbf{H}^{(k)} \in \mathbb{R}^{n \times d_k}$. Throughout the paper, we will use the terms GNN and MPNN interchangeably, and will generally consider the most widely used instance of GNNs, which are Graph Convolutional Networks (GCNs) [67] whose matrix update equation is given by:

$$\mathbf{H}^{(k)} = \sigma\big(\hat{\mathbf{A}}\mathbf{H}^{(k-1)}\mathbf{W}^{(k-1)}\big), \tag{3}$$

where $\hat{\mathbf{A}} = (\mathbf{D} + \mathbf{I})^{-1/2}(\mathbf{A} + \mathbf{I})(\mathbf{D} + \mathbf{I})^{-1/2}$ is the adjacency matrix with added self connections through the identity matrix $\mathbf{I}$, and $\sigma(\cdot)$ is a nonlinearity. Our analysis also applies to Graph Attention Networks (GATs) [101], where the fixed normalized adjacency is replaced by a learned adjacency matrix which dynamically modulates connectivity while preserving the key spectral properties used in our analysis.

## 2.2 Recurrent Neural Networks

A Recurrent Neural Network (RNN) is a function $g_{\boldsymbol{\theta}} : \mathbf{x} \mapsto \mathbf{y}$, where $\mathbf{x} = (\mathbf{x}^{(1)}, \mathbf{x}^{(2)}, \dots, \mathbf{x}^{(K)})$ and $\mathbf{y} = (\mathbf{y}^{(1)}, \mathbf{y}^{(2)}, \dots, \mathbf{y}^{(K)})$, where $\mathbf{x}^{(k)} \in \mathbb{R}^d$ is the input vector at time step $k$ and $\mathbf{y}^{(k)} \in \mathbb{R}^m$ is the output vector at time step $k$, and $\boldsymbol{\theta}$ are learnable parameters. RNNs are designed to handle sequential data by maintaining a hidden state $\mathbf{h}^{(k)} \in \mathbb{R}^{d_h}$ that captures information from previous time steps. This hidden state[2] allows the network to model sequential dependencies in the data. The update equations for the hidden state of the RNN are as follows:

$$\mathbf{h}^{(k)} = \sigma(\mathbf{W}_h \mathbf{h}^{(k-1)} + \mathbf{W}_x \mathbf{x}^{(k)}). \tag{4}$$

This type of approach has deep connections with ideas from dynamical systems [99] and chaotic systems [39]. These ideas have become more relevant in recent work [54, 81], where the nonlinearity in (4) is removed in the interest of parallelization and the ability to directly control the dynamics of the system through the eigenvalues of the state transition matrix $\mathbf{W}_h$. We note that these types of approaches are also popular in the *reservoir computing* literature [63], where the state transition matrix is left untrained and more emphasis is placed on the dynamics of the model.

## 2.3 The Vanishing and Exploding Gradient Problem

Both RNNs and GNNs are trained using the chain rule. One can backpropagate gradients w.r.t. the weights at $i^{\text{th}}$ layer of a $K$-layer GNN or RNN as

$$\frac{\partial \mathcal{L}}{\partial \boldsymbol{\theta}^{(i)}} = \frac{\partial \mathcal{L}}{\partial \mathbf{H}^{(K)}} \left( \prod_{k=i+1}^{K} \frac{\partial \mathbf{H}^{(k)}}{\partial \mathbf{H}^{(k-1)}} \right) \frac{\partial \mathbf{H}^{(i)}}{\partial \boldsymbol{\theta}^{(i)}}, \tag{5}$$

where matrix $\mathbf{H}^{(k)}$ in an RNN will contain a single state vector. As identified by [82], a major issue in training this type of models arises from the *product Jacobian*, given by:

$$\mathbf{J} = \prod_{k=i+1}^{K} \frac{\partial \mathbf{H}^{(k)}}{\partial \mathbf{H}^{(k-1)}} = \prod_{k=i+1}^{K} \mathbf{J}_k. \tag{6}$$

In general, we have that if $||\mathbf{J}_k||_2 \approx \lambda$ for all layers, then $||\mathbf{J}||_2 \leqslant \lambda^{K-i}$. This means that we require $\lambda \approx 1$ for gradients to neither explode nor vanish, a condition also known as *edge of chaos*.

## 2.4 Over-smoothing, Over-squashing, and Vanishing Gradients in GNNs

**Node Feature Collapse and Over-smoothing.** GNNs are known to not perform well at large depths [69]. This issue has been heavily linked with the issue of *over-smoothing* [18, 80], which describes the tendency of GNNs to produce *smoother* representations as more and more layers are added. In Section 4, we study this issue from the lens of vanishing gradients and show that **the performance degradation of GNNs has a much simpler explanation**: it occurs due to the norm-contracting nature of GNN updates, which is also intimately related to some notions of smoothing presented in the literature [90].[3]

---

[2]Note that we purposefully maintain the same notation for the hidden state as the one in the previous subsection for node features.

[3]We elaborate on the definition we use in the paper around over-smoothing as it related to the broader literature in Appendix E.

**Over-squashing.** Over-squashing [1, 100, 30, 7] was originally introduced as a *bottleneck* resulting from 'squashing' into node representations amounts of information that are growing potentially exponentially quickly due to the topology of the graph. It is often characterized by the quantity $\left\|\partial \mathbf{h}_u^{(K)}/\partial \mathbf{h}_v^{(0)}\right\|$ being low, implying that the final representation of node $u$ is not very sensitive to the initial representation at some other node $v$. While the relationship between over-squashing and vanishing gradients was hinted at by [30], in Section 5 we explore this relationship in detail by showing that **techniques aimed to mitigate vanishing gradients in sequence models help to mitigate over-squashing in GNNs**.

**Vanishing gradients.** Vanishing gradients have been extensively studied in RNNs [11, 61, 82], while this problem has been surprisingly mostly overlooked in the GNN community. For a detailed discussion on the relevant literature, we point the reader to the Appendix F. We simply highlight that there are works that have seen success in taking ideas from sequence modelling [91, 47, 102, 10, 66] or signal propagation [40, 95] and bridging them to GNNs, but they rarely have a detailed discussion on vanishing gradients. In Section 5, we show that **vanishing gradient mitigation techniques from RNNs seem to be very effective towards the mitigation of feature collapse and over-squashing in GNNs** and argue that the two communities have very aligned problems and goals.

## 3 Connecting Sequence and Graph Learning through State-Space Models

In this section, we study GNNs from a sequence model perspective. We show that the most common classes of GNNs are more prone to vanishing gradients than feedforward or recurrent networks due to the spectral contractive nature of the normalized adjacency matrix. We then propose GNN-SSMs, a state-space-model-inspired construction of a GNN that allows more direct control of the spectrum.

### 3.1 Similarities and differences between learning on sequences and graphs

The GNN architectures that first popularized deep learning on graphs [17, 29] were initially presented as a generalization of Convolutional Neural Networks (CNNs) to irregular domains. GCNs [67] subsequently restricted the architecture in [29] to a one-hop neighborhood. While this is still termed "convolutional" (due to weight sharing across nodes), the iterative process of aggregating information from each node's neighborhood can also be viewed as *recurrent-like* state updates.

If we consider an RNN unrolled over time, it forms a directed path graph feeding into a state node with a self-connection, making it a special case of a GNN. Conversely, node representations in GNNs can be stacked using matrix vectorization, allowing us to interpret GNN layer operations as iterative state updates. This connection suggests that the main difficulty faced by RNNs, namely the vanishing and exploding gradients problem [82], may likewise hinder the learning ability of GNNs. We note, however, that one key difference between RNNs and GNNs is that RNN memory *only* depends on how much information is dissipated by the model during the hidden state update, whereas GNNs normalize messages by the inverse node degree, which introduces an additional information dissipation step that we will explore in more detail in Section 5.

### 3.2 Graph convolutional and attentional models are prone to extreme gradient vanishing

Based on the previously introduced notion of stacking node representations using the matrix vectorization operation, we now analyze the gradient dynamics of GNN. In particular, we focus on the gradient propagation capabilities of graph convolutional and attentional models at initialization, given their widespread use in the literature. Specifically, we demonstrate that the singular values of the layer-wise Jacobian in these models form a highly contractive mapping, which prevents effective information propagation beyond a few layers. We formalize this claim in Lemma 3.1 and Theorem 3.2, and we refer the reader to Appendix A.1 for the corresponding proofs.

**Lemma 3.1** (Spectrum of layer-wise Jacobian's singular values). *Let* $\mathbf{H}^{(k)} = \tilde{\mathbf{A}}\,\mathbf{H}^{(k-1)}\,\mathbf{W}$ *be a linear GCN layer, where* $\tilde{\mathbf{A}}$ *has eigenvalues* $\{\lambda_1,\ldots,\lambda_n\}$ *and* $\mathbf{W}\mathbf{W}^T$ *has eigenvalues* $\{\mu_1,\ldots,\mu_{d_k}\}$. *Consider the layer-wise Jacobian* $\mathbf{J} = \partial\,\mathrm{vec}\big(\mathbf{H}^{(k)}\big)/\partial\,\mathrm{vec}\big(\mathbf{H}^{(k-1)}\big)$, *then the squared singular values of* $\mathbf{J}$ *are given by the set*

$$\big\{\lambda_i^2\,\mu_j \ \big| \ i = 1,\ldots,n, \ \ j = 1,\ldots,d_k\big\}.$$

**Theorem 3.2** (Jacobian singular-value distribution). *Assume the setting of Lemma 3.1, and let $\mathbf{W} \in \mathbb{R}^{d_{k-1} \times d_k}$ be initialized with i.i.d. $\mathcal{N}(0, \sigma^2)$ entries. Denote the squared singular values of the Jacobian by $\gamma_{i,j}$. Then, for sufficiently large $d_k$ the empirical eigenvalue distribution of $\mathbf{W}\mathbf{W}^T$ converges to the Marchenko-Pastur distribution. Then, the mean and variance of each $\gamma_{i,j}$ are*

$$\mathbb{E}\big[\gamma_{i,j}\big] = \lambda_i^2\,\sigma^2, \tag{7}$$

$$\mathrm{Var}\big[\gamma_{i,j}\big] = \lambda_i^4\,\sigma^4\,\frac{d_k}{d_{k-1}}. \tag{8}$$

Theorem 3.2 shows that the singular-value spectrum of the Jacobian is modulated by the squared spectrum of the normalized adjacency. Since $|\lambda_i| \leqslant 1$ for all eigenvalues of the normalized adjacency, the ability of GCNs to propagate gradients is in expectation worse than that of RNNs or MLPs. In particular, iterating these operations causes the majority of the spectrum to shrink to zero more quickly than in classical deep linear [93] or nonlinear [84] networks. Moreover, using sigmoidal activations and orthogonal weights will not push the singular-value spectrum to the edge of chaos as in [84], due to the additional contraction from the adjacency.

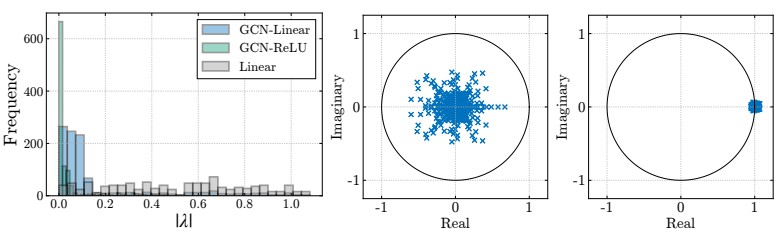

Figure 2: **Left:** Histogram of eigenvalue modulus of the Jacobian for linear, linear convolutional, and nonlinear convolutional layers. **Middle:** Vectorized Jacobian for GCN. **Right:** Vectorized Jacobian for GCN-SSM with $\mathrm{eig}(\Lambda) \approx 1$, $\mathrm{eig}(B) \approx 0.1$.

The effect of each operation on the layer-wise Jacobian is empirically demonstrated in Figure 2, which also showcases the contraction effect of the normalized adjacency. The figure reveals that even a single layer's Jacobian exhibits a long tail of squared singular values near zero. This spectral structure leads to ill-conditioned gradient propagation and non-isometric (not norm-preserving) signal dynamics. The same results hold for GATs, as the adjacency still exhibits a contractive spectral structure despite being learned during training.

Note that to overcome this contraction without altering the architecture, one would have to both set $\sigma^2$ in a way that precisely compensates for the normalized adjacency (which can be computationally expensive to estimate) and choose the nonlinearity carefully. In the next subsection, we present a general, simple, and computationally efficient method to place the Jacobian at the edge of chaos at initialization by writing feature updates in a state-space representation.

### 3.3 GNN-SSM: Improving the training dynamics of GNNs through state-space models

To allow direct control of the signal propagation dynamics of any GNN, we can rewrite its layer-to-layer update as a state-space model. Concretely, we express the update as

$$\mathbf{H}^{(k+1)\top} = \mathbf{\Lambda}\,\mathbf{H}^{(k)\top} + \mathbf{B}\,\mathbf{X}^{(k)\top}$$
$$= \mathbf{\Lambda}\,\mathbf{H}^{(k)\top} + \mathbf{B}\,\mathbf{F}_{\boldsymbol{\theta}}\big(\mathbf{H}^{(k)}, k\big)^{\top} \tag{9}$$

where we refer to $\mathbf{\Lambda}$ as the *state transition matrix* and $\mathbf{B}$ as the *input matrix*,[4] and $\mathbf{F}_{\boldsymbol{\theta}}(\mathbf{H}^{(k)}, k)$ as a time-varying *coupling function* which connects each node to some neighborhood. We refer to the model defined in Equation (9) as GNN-SSM. From an RNN perspective, $\mathbf{\Lambda}$ plays the role of the "memory", in charge of recalling all the representations at each layer at the readout layer, while the neighborhood aggregation plays the role of an input injected into the state via $\mathbf{B}$. In traditional GNNs, this recurrent memory mechanism is absent, so these models act in a *memoryless* way: features at one layer do not explicitly store or retrieve past information in the way a stateful model would.

In the state-space view, the eigenvalues of $\mathbf{\Lambda}$ determine the *memory dynamics*: large eigenvalues can preserve signals (or, if above unity, cause exploding modes), whereas small eigenvalues quickly

---

[4]Here, we deviate from the traditional state-space formalism, which uses $\mathbf{A}$ as the state transition matrix, since we use this notation for the adjacency. Further, we employ transposes to increase the resemblance to the traditional SSM updates.

attenuate them. Meanwhile, $\mathbf{B}$ controls which aspects of the node features get injected into the hidden state at each step. Because this framework is agnostic to the exact coupling function, any MPNN layer can serve as $\mathbf{F}_{\theta}$. We showcase the effect of these matrices on the layer-wise Jacobian in Proposition 3.3.

**Proposition 3.3** (Effect of state-space matrices). *Consider the setting in* (9) *and* $\Gamma = \partial \operatorname{vec}(\mathbf{F}_{\theta}(\mathbf{H}^{(k)}))/\partial \operatorname{vec}(\mathbf{H}^{(k)})$. *Let* $\otimes$ *denote the Kronecker product. Then, the norm of the vectorized Jacobian* $\mathbf{J}$ *is bounded as:*

$$\|\mathbf{J}\|_2 \leqslant \|I_{d_k} \otimes \mathbf{\Lambda}\|_2 + \|I_{d_k} \otimes \mathbf{B}\|_2 \|\Gamma\|_2$$
$$= \|\mathbf{\Lambda}\|_2 + \|\mathbf{B}\|_2 \|\Gamma\|_2, \tag{10}$$

The result above shows that the spectrum of the Jacobian is controlled through the eigenvalues of $\mathbf{\Lambda}$. If we have that the spectrum of $\Gamma$ is around zero, it suffices to have $\operatorname{eig}(\Lambda) \approx 1$ to bring the vectorized Jacobian to the edge of chaos. We empirically validate this in Figure 2.

For simplicity and clarity of conclusions, we consider $\mathbf{\Lambda}$ and $\mathbf{B}$ to be *shared across layers* and *fixed* (i.e., not trained by gradient descent) to guarantee the desired properties. To construct $\mathbf{\Lambda}$, we first generate a random unitary matrix, which has all eigenvalues on the unit circle, and then scale it to control the spectral radius[5], allowing us to design with precise control over the system dynamics. Only the coupling function $\mathbf{F}_{\theta}$ is optimized. Empirically, we observe that this simpler scheme actually improves downstream performance in some settings. We highlight, however, that this is the most simple instance of a more general framework that aims to incorporate ideas from recurrent processing into GNNs without losing permutation-equivariance. One could easily extend this state-space idea to include more complex gating [61] or other constraints on the state transition matrix [60].

# 4 How does Extreme Gradient Vanishing affect Over-smoothing?

In this section, we study the practical implications of the mechanism causing vanishing gradients in GNNs in relation to node feature evolution. We show empirically and theoretically how GNN layers acting as *contractions* make node features collapse to a fixed point. We experimentally validate our points by analyzing Dirichlet energy, node feature norms, and node classification performance for increasing numbers of layers. Overall, we believe this section provides a more *practical and general* understanding of the consequences of extreme vanishing in GNNs by analyzing them from the point of view of their layer-wise Jacobians.

## 4.1 A contractive GNN leads to node feature collapse

We consider in our analysis GNN layers as in Equation 3. We view a GNN layer as a map $f_k : \mathbb{R}^{nd} \to \mathbb{R}^{nd}$ and construct a deep GNN $f$ via composition of $K$ layers, i.e. $f = f_K \circ \cdots \circ f_1$. Let $\mathbf{J}_f \in \mathbb{R}^{nd \times nd}$ denote the layer-wise Jacobian of a GNN $f$. [6] The supremum of the Jacobian (if well-defined) of $f$ over a convex set $U$ corresponds to the Lipschitz constant $\|f\|_{\text{Lip}}$ [58], i.e. $\|f\|_{\text{Lip}} = \sup_{\mathbf{H} \in U} \|\mathbf{J}_f(\mathbf{H})\|$, where by submultiplicativity of Lipschitz constants we have that $\|f\|_{\text{Lip}} \leqslant \prod_{k=1}^{K} \|f_k\|_{\text{Lip}}$. We point to the Appendix A.2 for a more detailed explanation of the objects in question. A Lipschitz function $f$ is *contractive* if $\|f\|_{\text{Lip}} < 1$. We now assume that $\|f_k\|_{\text{Lip}} < 1$ for all $k$, meaning that each layer is a *contraction mapping*.[7]

**Lemma 4.1** (Banach Fixed Point Theorem [5]). *Let* $f$ *be an operator with Lipschitz constant* $\|f\|_{Lip} < 1$. *Then for any starting point* $\mathbf{x}_0$, *the fixed-point iteration* $\mathbf{x}_{n+1} = f(\mathbf{x}_n)$ *converges to the unique fixed point of* $f$ *at a linear rate* $O\left(1/\|f\|_{Lip}^n\right)$

From Lemma 4.1 above and the extreme gradient vanishing results presented in Section 3.2, we can distinguish two cases for contractive GNN layers: (1) With shared layers (as in [31]), repeated applications of the GNN will result in convergence to a unique fixed point; (2) with repeated application of non-shared but highly contracting layers, the overall GNN function will converge to, or very close to, a unique fixed point in a single forward pass due to the low Lipschitz constant of

---

[5]An alternative strategy consists in computing the matrix eigendecomposition, manually place the eigenvalues where desired, and reconstruct the matrix.

[6]In our analysis, it is important that the input to the GNN is a vector in $\mathbb{R}^{nd}$ rather than a matrix in $\mathbb{R}^{n \times d}$, as the Jacobians and norms are different for the two cases. For this reason, it is important to take care in the definitions of these objects.

[7]Note that the analysis holds for any submultiplicative matrix norm.

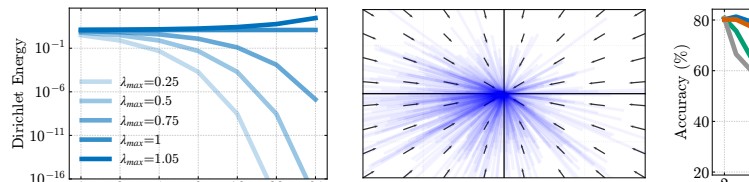

Figure 3: Experimental evaluation on Cora for an increasing number of layers. **Left:** Dirichlet Energy evolution for different $||\Lambda||_2$. **Middle:** 2-Dimensional random feature projection evolution with a fixed point at zero. **Right:** Node classification performance.

the overall GNN. Both cases result in convergence towards a fixed point as all nodes evolve using common transformations. In practice, we observe that node representations tend to collapse to a zero norm node state, see Figure 3. To study this further, we consider a GNN update, under the following assumption, which is consistent with the setup in (3):

**Lemma 4.2.** *Consider a GNN layer $f_K$ as in Equation 3, with non-linearity $\sigma$ such that $\sigma(0) = 0$ (e.g. ReLU or $\tanh$). Then, $f(\mathbf{0}) = \mathbf{0}$, i.e. $\mathbf{0}$ is a fixed point of $f$.*

Then, we have that node representations will evolve as in Proposition 4.3 presented next.

**Proposition 4.3** (Convergence to a unique fixed point.). *Let $\|f_k\|_{Lip} < 1 - \epsilon$ for some $\epsilon > 0$ for all $k = 1 \dots L$. Then, for $\mathbf{H} \in U \subseteq \mathbb{R}^{nd}$, we have that:*

$$\|f(\mathbf{H})\| < (1 - \epsilon)^K \|\mathbf{H}\| < \|\mathbf{H}\|. \tag{11}$$

*In particular, as $K \to \infty$, $f(\mathbf{H}) \to \mathbf{0}$.*

In other words, in the setting we have considered, if layers $f_k$ are contractive, their repeated application will monotonically converge to the *unique* fixed point $\mathbf{0}$, by Lemmas 4.2 and 4.1, which serves to explain the behavior observed empirically. We note that this is similar to the analysis in [88], but we highlight that our analysis is much more broad, as it applied to a number of models beyond GCNs, and is it only required knowledge of behavior of each layer through the Lipschitz constant. Furthermore, we emphasize the important connection between the Lipschitz constant and the vanishing gradients problem, which are linked through the Jacobian.

The collapse of node features to a single point has been typically described as *over-smoothing*. In particular, over-smoothing describes the tendency of node features to become too similar to each other as more layers are added in GNNs [18]. A common way of measuring over-smoothing in GNNs, see [91, 90], is via the *unnormalized Dirichlet energy $\mathcal{E}(\mathbf{H})$*.[8] Given a feature matrix $\mathbf{H} \in \mathbb{R}^{n \times d}$ on an unweighted graph $G$, $\mathcal{E}(\mathbf{H})$ takes the form:

$$\mathcal{E}(\mathbf{H}) = \text{tr}\left(\mathbf{H}^\top \mathbf{\Delta} \mathbf{H}\right) = \sum_{(u,v) \in E} \|\mathbf{h}_u - \mathbf{h}_v\|^2, \tag{12}$$

where $\mathbf{\Delta}$ is the unnormalized graph Laplacian [28]. The Dirichlet energy measures the *smoothness* of a signal over a graph and will be minimized when the signal is constant over each node – at least when using the unnormalized Laplacian. In Proposition 4.4, we show how the layer-wise Jacobians relate to the unnormalized Dirichlet energy.

**Proposition 4.4** (Contractions decrease Dirichlet energy.). *Let $f$ be a GNN, $|E|$ be the number of edges in $G$, and $\mathbf{H} \in \mathbb{R}^{nd}$. We have the following bound:*

$$\mathcal{E}(f(\mathbf{H})) \leqslant 2|E| \prod_{k=1}^{K} \|f_k\|_{Lip}^2 \|\mathbf{H}\|^2. \tag{13}$$

*In particular, if $\|f_k\|_{Lip} < 1 - \epsilon$ for some $\epsilon > 0$ for all $k = 1 \dots K$, then as $K \to \infty$ we have that $\mathcal{E}(f(\mathbf{H})) \to 0$.*

This result shows that the energy is directly controlled by the norm of the input signal $\mathbf{H}$ and by the contracting effect of the layers $f_k$. The repeated application of contractive layers results in the unnormalized Dirichlet energy being artificially lowered as signals are gradually reduced in norm.

---

[8]We offer a more precise characterization oh how this relates to the normalized Dirichlet energy of node features in Appendix E.

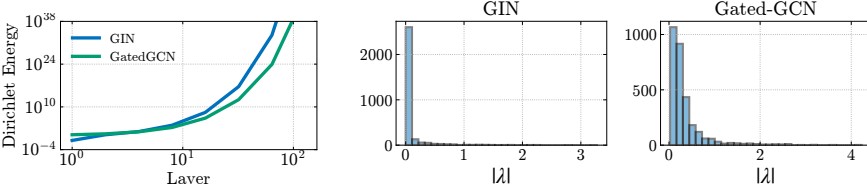

Figure 4: **Left:** Evolution of Dirichlet Energy on the Cora dataset for GIN and Gated-GCN. **Right:** Histograms of eigenvalue spectra of layer-to-layer Jacobians for GIN and Gated-GCN.

**Important consequences of our theoretical results.** **1)** The most important takeaway of the analysis above is that *vanishing gradients are directly connected to over-smoothing through the Lipschitz constant*. In particular, the same mechanism that causes gradient vanishing issues, is responsible for the collapse of all features to a **unique zero fixed point** (i.e. zero feature collapse) where the Dirichlet energy is minimized. The quick collapse of traditional graph convolutional and attentional models can also be understood from the extreme gradient vanishing result introduced in Section 3. As such, while much of the literature, starting with [80, 18], has attributed over-smoothing to iterative aggregation and convergence to a rank-one subspace, our analysis shows that this phenomenon is instead an artifact of representations collapsing to zero. As later noted by [3] (without any formal argument), over-smoothing is not necessarily the cause of performance degradation. **Here, we provide a theoretical explanation for the poor performance of MPNNs at large depths, linking it to inherent trainability limitations arising from vanishing (and in some cases exploding) gradients.**

**2)** This result also provides a connection between the study of GNNs and the study of signal propagation (or dynamical isometry) in feedforward networks [93, 86, 84] and recurrent neural networks [61, 2, 81]. In the dynamical isometry literature, the primary interest is to improve the learning times of deep feedforward networks, whereas the recurrent neural network literature focuses on memory and long-range information retrieval. We highlight that *connecting ideas from these fields of study will enable the design of new models that benefit from both worlds*, even though these techniques were originally developed with different objectives in mind. This also serves as an explanation for why simple modifications, such as **residual connections or normalization**, worked in practice to mitigate over-smoothing, given their links to dynamical isometry [107, 75].

**3)** Finally, we highlight that this result provides an objective *evaluation metric* to gauge whether a GNN will over-smooth or not. We hope that the eigenanalysis of the Jacobian will become a widespread empirical test used for this purpose.

### 4.2 Empirical validation of theoretical results

To validate the theory above, we perform a series of empirical tests. In particular, we check the evolution of the Dirichlet energy, latent vector norms, and node classification accuracy on the Cora dataset as the number of layers of different models is increased. The results are presented in Figure 3. Further, we present additional experiments for different graph structures and models in Appendix D.1, and showcase how several models with edge-of-chaos Jacobians result in no over-smoothing, reinforcing the generality of this result beyond classical MPNNs.

From Figure 3, we see that one can exactly control the evolution of the Dirichlet energy of the system through the spectrum of the Jacobian, which can, in turn, be modified through the spectrum of $\Lambda$ in the GNN-SSM model. Furthermore, this shrinks faster the lower the norm of the Jacobian is, which validates Proposition 4.4. Beyond a Dirichlet energy analysis of the system, notice that node classification performance does not deteriorate when $\text{eig}(\Lambda) \approx 1$, and improves over simply applying an SSM layer with no modulation of the hyperparameters or a residual connection without gating. The dynamics of the GNN in this setting are shown in Figure 1.[9]

**Results for Additional MPNNs.** In this section, we have focused on GCN and GAT because their layer operators admit clean characterizations, which makes the analysis mathematically tractable. However, the arguments extend to *any* GNN, where it suffices to examine the eigenvalues of the *layer-*

---

[9]Note that in the case of GCN, it is sufficient to scale weight norms to prevent the Dirichlet energy from collapsing to zero with additional layers, as the Dirichlet energy only depends on the maximum eigenvalue of the Jacobian. However, obtaining good performance on node-classification tasks required to transport the whole spectrum to be centred around the edge of chaos to obtain good performance. We elaborate on this in App. E

*wise Jacobians.* To make this concrete, in Figure 4 we plot (i) histograms of the Jacobian eigenvalues for GIN [106], and Gated-GCN [16], and (ii) the evolution of each model's Dirichlet energy as the number of layers increases. We observe that GIN exhibits occasional outliers with $|\lambda| > 1$ and Gated-GCN displays a sizeable mass above the unit circle, which indicates unstable dynamics. This perspective helps explain several empirical findings reported in [3]: the divergence of Dirichlet energy in some architectures follows directly from norm expansion driven by unstable Jacobians, whereas the zero feature collapse seen in GCN and GAT reflects their contraction. Consequently, while *not all* GNNs oversmooth (some instead become unstable and blow up) **the generally poor performance of many GNNs at large depths can be understood through their learning dynamics: collapse under contraction (as in GCNs and GATs, especially with ReLUs) or divergence under instability.**

## 5 The Impact of Vanishing Gradients on Over-squashing

In this section, we study the connection between vanishing gradients and over-squashing in GNNs, which fills in the gaps and open questions left in the analysis of [30]. [10]

### 5.1 Mitigating over-squashing by combining increased connectivity and non-dissipativity

Over-squashing is typically measured via the sensitivity of a node embedding after $k$ layers with respect to the input of another node using the node-wise Jacobian.

**Theorem 5.1** (Sensitivity bounds, [30]). *Consider a standard MPNN with $k$ layers, where $c_\sigma$ is the Lipschitz constant of the activation $\sigma$, $w$ is the maximal entry-value over all weight matrices, and $d$ is the embedding dimension. For $u, v \in V$ we have*

$$\left\| \frac{\partial \mathbf{h}_v^{(k)}}{\partial \mathbf{h}_u^{(0)}} \right\| \leqslant \underbrace{(c_\sigma w d)^k}_{model} \underbrace{(\mathbf{O}^k)_{vu}}_{topology}, \tag{14}$$

*where $\mathbf{O} = c_r \mathbf{I} + c_a \mathbf{A} \in \mathbb{R}^{n \times n}$ is the message passing matrix adopted by the MPNN, and where $c_r$ and $c_a$ are the contributions of the self-connection and aggregation term.*

Theorem 5.1 shows that the sensitivity of the node embedding is a combination of (i) a term based on the graph topology and (ii) a term dependent on the model dynamics, with over-squashing occurring when the right-hand side of Equation (14) becomes too small. We highlight that this differs from the standard product Jacobian, which arises in RNNs. This is because, in MPNNs, messages are scaled by the inverse node degree, incurring an extra information dissipation step. Consequently, while recurrent architectures only need to adjust their dynamics to ensure long memory, MPNNs must *simultaneously* enhance graph connectivity and modify their dynamics to mitigate vanishing and exploding gradients.

Even though the sensitivity bound in Theorem 5.1 is controlled by two components, the majority of the literature has typically focused on addressing only the topological term via *graph rewiring* [42, 100, 65, 8, 41], with some methods also targeting the model dynamics [47, 48, 59]. In fact, [30] explicitly discourages increasing the model term in Theorem 5.1 and claims that doing so could lead to over-fitting and poorer generalization. However, we argue that increasing the model term, directly linked to vanishing gradients as discussed in Section 4, is essential to mitigate over-squashing. Rather than harming performance, boosting this term helps prevent over-smoothing, since even in a well-connected graph where information can be reached in fewer hops, unaddressed vanishing gradients due to the model term will cause the target node's features to collapse during message passing. Frameworks combining these strategies include [55], which integrates graph rewiring with a delay term, and [32], which

Table 1: Ablation on LRGB datasets. $d \uparrow$ adds latent dims; $-$ removes; $+$ adds.

| **Model** | Pept-func AP↑ | Pept-struct MAE↓ ($\times 10^{-2}$) |
|---|---|---|
| GCN | $60.93_{\pm 0.138}$ | $33.41_{\pm 0.041}$ |
| kGCN-SSM | $\underline{69.02}_{\pm 0.218}$ | $28.98_{\pm 0.324}$ |
| $+d \uparrow$ | $\mathbf{72.12}_{\pm 0.268}$ | $27.01_{\pm 0.071}$ |
| $-\text{eig}(\Lambda) \approx 1$ | $61.41_{\pm 0.724}$ | $\mathbf{25.81}_{\pm 0.032}$ |
| $-$SSM | $57.76_{\pm 1.971}$ | $\underline{26.02}_{\pm 0.213}$ |
| $-$khop | $60.93_{\pm 0.138}$ | $33.41_{\pm 0.041}$ |
| DRew-GCN | $68.04_{\pm 1.442}$ | $27.66_{\pm 0.187}$ |
| $+d \uparrow$ | $68.05_{\pm 0.626}$ | $27.64_{\pm 0.067}$ |
| $-$Delay | $49.02_{\pm 2.512}$ | $27.08_{\pm 0.041}$ |

---

[10]In this part, we mostly focus on long-range interactions, which has been in many cases treated synonymously with over-squashing in the literature. For a more precise characterization of the relationship between over-squashing and long range, we point the reader to [3]. We highlight that we believe the distinction between computational and topological bottlenecks is not particularly relevant in our setting.

merges multi-hop aggregation with ideas from SSMs.[11] These approaches have generally led to state-of-the-art results, significantly improving performance over standalone rewiring.

## 5.2 Empirical validation of claims

We focus our empirical validation on answering the following questions: (i) What is the result of combining an effective rewiring scheme with vanishing gradient mitigation? (ii) Will this result in similar state-of-the-art results? To investigate this, we construct a minimal model that combines high connectivity with non-dissipativity. In particular, we make of the GNN-SSM model and employ a k-hop aggregation scheme for the coupling function $\mathbf{F}_\theta$, which we term kGNN-SSM (more details are provided in Appendix B).

We start by testing the performance on the RingTransfer task introduced in [30], as it is a task where we certifiably know that long-range dependencies exist. We modify the eig($\Lambda$) in the kGNN-SSM to move the Jacobian from the edge of stability to a progressively more dissipative state. The results are shown in Figure 5. From the figure, we see that (i) kGNN-SSM achieves state-of-the art performance only when coupling strong connectivity and an edge of chaos Jacobian (ii) making the model more dissipative directly results in worse

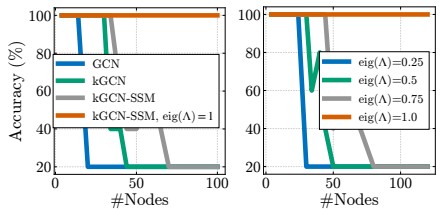

Figure 5: **Left:** Performance on the RingTransfer task. **Right:** Effect of dissipativity.

long-range modeling capabilities. We believe the latter point demonstrates the importance of the model term in Theorem 5.1.

Table 2: Mean and std. for test $log_{10}(\text{MSE})$ averaged over 4 random weight initializations on the GPP tasks

| Model | Diam. | SSSP | Ecc. |
|---|---|---|---|
| GCN | $0.742_{\pm 0.047}$ | $0.950_{\pm 9.18 \cdot 10^{-5}}$ | $0.847_{\pm 0.003}$ |
| + SSM | $-2.431_{\pm 0.033}$ | $-2.821_{\pm 0.565}$ | $-2.245_{\pm 0.003}$ |
| + eig($\Lambda$) $\approx 1$ | $\underline{-2.444}_{\pm 0.098}$ | $\underline{-3.593}_{\pm 0.103}$ | $\underline{-2.258}_{\pm 0.009}$ |
| + k-hop | $\mathbf{-3.075}_{\pm 0.055}$ | $\mathbf{-3.604}_{\pm 0.029}$ | $\mathbf{-4.265}_{\pm 0.178}$ |
| DRew-GCN | $-2.369_{\pm 0.105}$ | $-1.591_{\pm 0.003}$ | $-2.100_{\pm 0.026}$ |
| + delay | $-2.402_{\pm 0.110}$ | $-1.602_{\pm 0.008}$ | $-2.029_{\pm 0.024}$ |

Next, we ablate each component of the model on three graph property prediction tasks introduced in [47] alongside the real-world long-range graph benchmark (LRGB) from [36], focusing on the peptides-func and peptides-struct tasks. Additional details regarding the datasets and the experimental setting are reported in Appendix C. Here, we focus on ablating the effect of rewiring, adding an SSM layer, and placing the model at the edge of chaos through $\mathbf{\Lambda}$. In the LRBG tasks, we additionally ablate the effect of increasing the hidden memory size, as we consider forty layers in the peptides-func dataset, which requires more long-range capabilities. Here, we also ablate DRew [55] under the same settings. We also provide a more detailed comparison with other models in Appendix D.3, and provide additional comments around the LRGB tasks in Appendix D.4.

The results are shown in Tables 1 and 2. Across the board, we observe that kGNN-SSM not only matches DRew-Delay, but also outperforms it by a large amount in all cases, showcasing the strength of our state-space approach. In particular, we generally observe significant decreases in performance when removing both the high connectivity and non-dissipativity components of the model, highlighting their individual importance. Finally, we see that increasing memory size plays a big role in the peptides-func task, which is in line with observations made in the sequence modeling literature [53].

## 6 Conclusion

In this work, we revisit the well-known problems of representational collapse and over-squashing in GNNs from the lens of *vanishing gradients*, by studying GNNs from the perspective of recurrent and state-space models. In particular, we show that GNNs are prone to a phenomenon we term *extreme gradient vanishing*, which results in ill-conditioned signal propagation with few layers. As such, we argue that it is important to control the layerwise Jacobian and propose a state-space-inspired GNN model, termed GNN-SSM, to do so. We then uncover that vanishing gradients result in a *specific* form of over-smoothing in which all signals converge exactly to a unique fixed point, and support this claim empirically. Finally, we theoretically argue and empirically show that the mitigation of over-squashing is best achieved through a combination of strong graph connectivity and non-dissipative dynamics.

---

[11]Further links between the delay term and vanishing gradients are discussed in Appendix D.2. Further, we show that models tend to converge to the edge of chaos during training in Appendix D.3.

## Acknowledgments and Disclosure of Funding

AA and XD thank the Oxford-Man Institute for financial support. XD also acknowledges support from EPSRC No. EP/T023333/1. AA thanks T. Anderson Keller for engaging comments on early versions of the manuscript, Max Welling for valuable pointers to the dynamical isometry literature. AG and CG acknowledge funding from EU-EIC EMERGE (Grant No. 101070918). CG acknowledges support from NEURONE, a project funded by the European Union - Next Generation EU, M4C1 CUP I53D23003600006, under program PRIN 2022 (prj. code 20229JRTZA).

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

# A  Theoretical Results

## A.1  Proofs of Jacobian Theorems

**Definition A.1** (Vectorization and Kronecker product). Let $\mathbf{X} \in \mathbb{R}^{m \times n}$ be a real matrix. The *vectorization* of $\mathbf{X}$, denoted $\mathrm{vec}(\mathbf{X})$, is the $(mn)$-dimensional column vector obtained by stacking the columns of $\mathbf{X}$:

$$\mathrm{vec}(\mathbf{X}) = \begin{bmatrix} \mathbf{X}_{:,1} \\ \mathbf{X}_{:,2} \\ \vdots \\ \mathbf{X}_{:,n} \end{bmatrix} \in \mathbb{R}^{mn}.$$

One key property of the vectorization operator is its relationship to the Kronecker product. In particular, for compatible matrices $\mathbf{A}, \mathbf{B}, \mathbf{C}$, we have

$$\mathrm{vec}(\mathbf{A}\,\mathbf{B}\,\mathbf{C}) = (\mathbf{C}^T \otimes \mathbf{A})\,\mathrm{vec}(\mathbf{B}).$$

Here, $\otimes$ denotes the Kronecker product.

**Definition A.2** (Wishart matrix). Let $\mathbf{X} \in \mathbb{R}^{n \times p}$ be a matrix with i.i.d. entries $X_{ij} \sim \mathcal{N}(0, \sigma^2)$. The random matrix $\mathbf{X}^T \mathbf{X} \in \mathbb{R}^{p \times p}$ is called a *Wishart matrix* (up to a scaling factor). In particular, such a matrix follows the Wishart distribution $\mathcal{W}_p(n, \sigma^2)$ in certain parametrizations.

**Definition A.3** (Marchenko–Pastur distribution. [72]). In the high-dimensional limit ($n, p \to \infty$ at a fixed ratio $p/n \to c$), the empirical eigenvalue distribution of the (properly normalized) Wishart matrix $\mathbf{X}^T \mathbf{X}$ converges to the *Marchenko–Pastur distribution*. Concretely, if $\mathbf{X} \in \mathbb{R}^{n \times p}$ has entries $\mathcal{N}(0,1)$, then the eigenvalues of $\mathbf{X}^T \mathbf{X}$ lie within $[(1 - \sqrt{c})^2, (1 + \sqrt{c})^2]$ for large $n, p$, and their density converges to

$$f_{\mathrm{MP}}(x) = \frac{1}{2\pi c\, x}\sqrt{(x - a_{\min})(a_{\max} - x)}, \quad x \in [a_{\min}, a_{\max}],$$

with $a_{\min} = (1 - \sqrt{c})^2$ and $a_{\max} = (1 + \sqrt{c})^2$. If the entries of $\mathbf{X}$ have variance $\sigma^2 \neq 1$, then the support is rescaled by $\sigma^2$.

**Lemma A.4** (Spectrum of the Jacobian's singular values). *Let* $\mathbf{H}^{(k)} = \tilde{\mathbf{A}}\,\mathbf{H}^{(k-1)}\,\mathbf{W}$ *be a linear GCN layer, where* $\tilde{\mathbf{A}}$ *has eigenvalues* $\{\lambda_1, \dots, \lambda_n\}$ *and* $\mathbf{W}\mathbf{W}^T$ *has eigenvalues* $\{\mu_1, \dots, \mu_{d_k}\}$. *Consider the layer-wise Jacobian* $\mathbf{J} = \partial\,\mathrm{vec}(\mathbf{H}^{(k)})/\partial\,\mathrm{vec}(\mathbf{H}^{(k-1)})$, *Then the squared singular values of* $\mathbf{J}$ *are given by the set*

$$\big\{ \lambda_i^2\,\mu_j \;\big|\; i = 1, \dots, n, \;\; j = 1, \dots, d_k \big\}.$$

*Proof.* By the property of vectorization (Definition A.1), we have

$$\mathrm{vec}(\tilde{\mathbf{A}}\,\mathbf{H}^{(k-1)}\,\mathbf{W}) = (\mathbf{W}^T \otimes \tilde{\mathbf{A}})\,\mathrm{vec}(\mathbf{H}^{(k-1)}).$$

Hence

$$\mathbf{J} = \mathbf{W}^T \otimes \tilde{\mathbf{A}}.$$

By properties of the Kronecker product, the eigenvalues of $\mathbf{J}\,\mathbf{J}^T$ are the products of the eigenvalues of $\mathbf{W}^T \mathbf{W}$ and $\tilde{\mathbf{A}}^2$. Equivalently,

$$\mathrm{spec}(\mathbf{J}\,\mathbf{J}^T) = \mathrm{spec}(\mathbf{W}^T \mathbf{W}) \;\otimes\; \mathrm{spec}(\tilde{\mathbf{A}}^2),$$

where spec is the vectorized version of the set of eigenvalues of a matrix. If $\mathbf{W}^T \mathbf{W}$ has eigenvalues $\{\mu_j\}_{j=1}^{d_k}$ and $\tilde{\mathbf{A}}^2$ has eigenvalues $\{\lambda_i^2\}_{i=1}^{n}$, then the squared singular values of $\mathbf{J}$ are precisely $\lambda_i^2\,\mu_j$ for $i \in \{1, \dots, n\}, j \in \{1, \dots, d_k\}$. $\qquad\square$

**Theorem A.5** (Jacobian singular-value distribution). *Assume the setting of Lemma 3.1, and let* $\mathbf{W} \in \mathbb{R}^{d_{k-1} \times d_k}$ *be initialized with i.i.d.* $\mathcal{N}(0, \sigma^2)$ *entries. Denote the squared singular values of the Jacobian by* $\gamma_{i,j}$. *Then, for sufficiently large* $d_k$ *the empirical eigenvalue distribution* $\mathbf{W}\mathbf{W}^T$ *converges to the Marchenko-Pastur distribution. Then, the mean and variance of each* $\gamma_{i,j}$ *are*

$$\mathbb{E}[\gamma_{i,j}] = \lambda_i^2\,\sigma^2, \tag{15}$$

$$\mathrm{Var}[\gamma_{i,j}] = \lambda_i^4\,\sigma^4\,\frac{d_k}{d_{k-1}}. \tag{16}$$

*Proof.* In this setting, $\mathbf{W}\mathbf{W}^T$ is Wishart if $\mathbf{W}$ has i.i.d. Gaussian entries. Its eigenvalues $\mu_j$ thus converge to the Marchenko–Pastur distribution for large $d_k$. From standard results on the moments of Wishart eigenvalues,

$$\mathbb{E}(\mu_j) \;=\; \sigma^2, \quad \mathrm{Var}(\mu_j) \;=\; \sigma^4 \,\frac{d_k}{d_{k-1}}.$$

Since $\gamma_{i,j} = \lambda_i^2 \,\mu_j$, we obtain

$$\mathbb{E}[\gamma_{i,j}] \;=\; \lambda_i^2 \,\mathbb{E}[\mu_j] \;=\; \lambda_i^2 \,\sigma^2,$$

$$\mathrm{Var}[\gamma_{i,j}] \;=\; \lambda_i^4 \,\mathrm{Var}(\mu_j) \;=\; \lambda_i^4 \,\sigma^4 \,\frac{d_k}{d_{k-1}}.$$

This completes the proof. $\qquad\square$

**Proposition A.6** (Effect of state-space matrices). *Consider the setting in* (9) *and* $\Gamma = \partial \, \mathrm{vec}(\mathbf{F}_{\boldsymbol{\theta}}(\mathbf{H}^{(k)}))/\partial \, \mathrm{vec}(\mathbf{H}^{(k)})$. *Let $\otimes$ denote the Kronecker product. Then, the norm of the vectorized Jacobian $\mathbf{J}$ is bounded as:*

$$\begin{aligned}
\|\mathbf{J}\|_2 &\leqslant \|I_{d_k} \otimes \mathbf{\Lambda}\|_2 + \|I_{d_k} \otimes \mathbf{B}\|_2 \|\Gamma\|_2 \\
&= \|\mathbf{\Lambda}\|_2 + \|\mathbf{B}\|_2 \|\Gamma\|_2,
\end{aligned} \tag{17}$$

*Proof.* We start by writing

$$\mathbf{J} \;=\; (I_{d_k} \otimes \mathbf{\Lambda}) \;+\; (I_{d_k} \otimes \mathbf{B})\,\Gamma.$$

Using the triangle inequality for the spectral norm,

$$\|\mathbf{J}\|_2 \;=\; \left\|(I_{d_k} \otimes \mathbf{\Lambda}) \;+\; (I_{d_k} \otimes \mathbf{B})\,\Gamma\right\|_2 \;\leqslant\; \|I_{d_k} \otimes \mathbf{\Lambda}\|_2 \;+\; \|(I_{d_k} \otimes \mathbf{B})\,\Gamma\|_2.$$

By the submultiplicative property of the spectral norm,

$$\|(I_{d_k} \otimes \mathbf{B})\,\Gamma\|_2 \;\leqslant\; \|I_{d_k} \otimes \mathbf{B}\|_2 \,\|\Gamma\|_2.$$

Since $\|I_{d_k} \otimes \mathbf{M}\|_2 = \|\mathbf{M}\|_2$ for any matrix $\mathbf{M}$, we obtain

$$\|I_{d_k} \otimes \mathbf{\Lambda}\|_2 \;=\; \|\mathbf{\Lambda}\|_2 \quad\text{and}\quad \|I_{d_k} \otimes \mathbf{B}\|_2 \;=\; \|\mathbf{B}\|_2.$$

Hence,

$$\|\mathbf{J}\|_2 \;\leqslant\; \|\mathbf{\Lambda}\|_2 \;+\; \|\mathbf{B}\|_2 \,\|\Gamma\|_2.$$

$\qquad\square$

## A.2 Proofs to Smoothing Theorems

**Definition A.7** (Lipschitz continuity). A function $f : \mathbb{R}^n \to \mathbb{R}^m$ is Lipschitz continuous if there exists an $L \geqslant 0$ such that for all $\mathbf{x}, \mathbf{y} \in \mathbb{R}^n$, we have that:

$$\|f(\mathbf{x}) - f(\mathbf{y})\| \;\leqslant\; L \,\|\mathbf{x} - \mathbf{y}\|,$$

where we equip $\mathbb{R}^n$ and $\mathbb{R}^m$ with their respective norms. The minimal such $L$ is called the Lipschitz constant of $f$.

The notion of Lipschitz continuity is effectively a bound on the rate of change of a function. It is therefore not surprising that one can relate the Lipschitz constant to the Jacobian of $f$. In particular, we state a useful and well-known result [58] that relates the (continuous) Jacobian map $\mathbf{J}_f$ of a continuous function $f : \mathbb{R}^n \to \mathbb{R}^m$ to its Lipschitz constant $L \geqslant 0$. In particular, the Lipschitz constant is is the supremum of the (induced) norm of the Jacobian taken over its domain.

**Lemma A.8** ([58]). *Let $f : \mathbb{R}^n \to \mathbb{R}^m$ be continuous, with continuous Jacobian $\mathbf{J}_f$. Consider a convex set $U \subseteq \mathcal{R}^n$ If there exists $L \geqslant 0$ such that $\|\mathbf{J}_f(\mathbf{x})\| \leqslant L$ for all $\mathbf{x} \in U$, then $\|f(\mathbf{x}) - f(\mathbf{y})\| \leqslant L \|\mathbf{x} - \mathbf{y}\|$. In particular, we have that the Lipschitz constant of $f$ $L$ is:*

$$L = \sup_{\mathbf{x}\in U} \|\mathbf{J}_f(\mathbf{x})\|.$$

The condition of $U$ being convex is a technicality that is easily achieved in practice with the assumption that input features are bounded and that therefore they live in a convex hull $U$. In particular, at each layer $k$ one can also find a convex hull $U_k$ such that the image of the layer $k-1$ is contained within $U_k$. We highlight that for non-linearities such as ReLU, there are technical difficulties when taking this supremum as there is a non-differentiable point at 0. This can be circumvented by considering instead a supremum of the (Clarke) generalized Jacobian [64]. We ignore this small detail in this work for simplicity as for ReLU this is equivalent to considering the supremum over $U/\mathbf{0}$, i.e. simply ignoring the problematic point $\mathbf{0}$.

**Lemma A.9.** *Consider a GNN layer $f_\ell$ as in Equation 3, with non-linearity $\sigma$ such that $\sigma(0) = 0$ (e.g. ReLU or $\tanh$). Then, $f(\mathbf{0}) = \mathbf{0}$, i.e. $\mathbf{0}$ is a fixed point of $f$.*

*Proof.* $f_\ell(\mathbf{0}) = \sigma\left(\hat{\mathbf{A}}\mathbf{0}\mathbf{W}\right) = \sigma(\mathbf{0}) = \mathbf{0}$. $\qquad\square$

**Proposition A.10** (Convergence to unique fixed point.)**.** *Let $\|f_\ell\|_{Lip} \leq 1 - \epsilon$ for some $\epsilon > 0$ for all $\ell = 1 \dots L$. Then, for $\mathbf{H} \in U \subseteq \mathbb{R}^{nd}$, we have that:*

$$\|f(\mathbf{H})\| \leq (1-\epsilon)^L \|\mathbf{H}\| < \|\mathbf{H}\|. \tag{18}$$

*In particular, as $L \to \infty$, $f(\mathbf{H}) \to \mathbf{0}$.*

*Proof.* By Lipschitz regularity of $f$ over $U$, we have that $\|f(\mathbf{x}) - f(\mathbf{y})\| \leq \|f\|_{\mathrm{Lip}} \|\mathbf{x} - \mathbf{y}\|$. Recall that by Lemma 4.2, we have that $f(\mathbf{0}) = \mathbf{0}$. This implies:

$$\begin{aligned}
\|f(\mathbf{H}) - f(\mathbf{0})\| = \|f(\mathbf{H})\| & \\
&\leq \|f\|_{\mathrm{Lip}} \|\mathbf{H}\| \\
&\leq \prod_{\ell=1}^{L} \|f_\ell\|_{\mathrm{Lip}} \|\mathbf{H}\| \\
&< \|\mathbf{H}\|,
\end{aligned}$$

where in the last step we use the fact that Lipschitz constants are submultiplicative and that for all $\ell$ we have that $\|f_\ell\|_{\mathrm{Lip}} < 1$ by assumption. The final statement is immediate by the Banach fixed point theorem and by noting that $f_\ell$ all share the same fixed point $\mathbf{0}$ by Lemma 4.2. $\qquad\square$

**Proposition A.11** (Contractions decrease Dirichlet energy.)**.** *Let $f$ be a GNN, $|E|$ be the number of edges in $G$, and $\mathbf{H} \in \mathbb{R}^{nd}$. We have the following bound:*

$$\mathcal{E}(f(\mathbf{H})) \leq 2|E| \prod_{\ell=1}^{L} \|f_\ell\|_{Lip}^2 \|\mathbf{H}\|^2. \tag{19}$$

*In particular, if $\|f_\ell\|_{Lip} \leq 1 - \epsilon$ for some $\epsilon > 0$ for all $\ell = 1 \dots L$, then as $L \to \infty$, $\mathcal{E}(f(\mathbf{H})) \to 0$.*

*Proof.* We denote by $f(\mathbf{H})|_i \in \mathbb{R}^d$, the $d$-dimensional evaluation of f(H) at node $i$. We make use of the inequality $\|f(\mathbf{H})|_i\| \leq \|\mathbf{H}\|$.

$$\mathcal{E}(f(\mathbf{H})) = \sum_{i \sim j} \| f(\mathbf{H})|_i - f(\mathbf{H})|_j \|^2$$
$$\leqslant \sum_{i \sim j} \| f(\mathbf{H})|_i \|^2 + \| f(\mathbf{H})|_j \|^2$$
$$\leqslant 2 \sum_{i \sim j} \| f(\mathbf{H}) \|^2$$
$$\leqslant 2 \| f \|_{\text{Lip}}^2 \sum_{i \sim j} \| \mathbf{H} \|^2$$
$$= 2 \| f \|_{\text{Lip}}^2 |E| \| \mathbf{H} \|^2$$
$$\leqslant 2 \prod_{\ell=1}^{L} \| f_\ell \|_{\text{Lip}}^2 |E| \| \mathbf{H} \|^2 .$$

It is then clear that, if $\| f_\ell \|_{\text{Lip}} \leqslant 1 - \epsilon$ for some $\epsilon > 0$ for all $\ell = 1 \dots L$, $\prod_{\ell=1}^{L} \| f_\ell \|_{\text{Lip}}^2 \leqslant (1 - \epsilon)^{2L} \to 0$ as $L \to \infty$.

We note that a similar procedure was used in [80, 18] for the specific case of GCNs. Our procedure is more general, as we use the Lipschitz constant of the network, which only requires knowledge of the input-output Jacobian of each layer of the network. In the case of GCN, this would encapsulate the dynamics of the adjacency and weight matrix, and also allows us to understand how any GNN (no matter how complex its internal structure) affects the Dirichlet energy, without requiring the use of heavy assumptions or simplifications for mathematical tractability. □

## B  kGNN-SSM: A simple method to combine high connectivity and non-dissipativity.

To test our assumption on more complex downstream tasks, we construct a minimal model that combines high connectivity with non-dissipativity. To guarantee high connectivity, we employ a k-hop aggregation scheme. In particular, each node $i$ at layer $k$ will aggregate information as

$$a_{i,k}^{(k)} = \psi^k \Big( \{ h_j^{(k)} : j \in \mathcal{N}_k(i) \} \Big), \tag{20}$$

where

$$\mathcal{N}_k(i) := \{ j \in V : d_G(i, j) = k \}$$

and $d_G : V \times V \to \mathbb{R}_{\geqslant 0}$ is the length of the minimal walk connecting nodes $i$ and $j$. This approach avoids a large amount of information being squashed into a single vector, and is more in line with the recurrent paradigm. We note that this scheme is similar to [32], but in this case we do not consider different block or parameter sharing, and our recurrent mechanism is based on an untrained SSM layer.

We denote a GNN endowed with this rewiring scheme and wrapped with our SSM layer as `kGNN-SSM`.

## C  Experimental Details

In this section, we provide additional experimental details, including dataset and experimental setting description and employed hyperparameters.

**Over-smoothing task.**   In this task, we aim to analyze the dynamics of the Dirichlet energy across three different graph topologies: Cora [108], Texas [83], and a grid graph. The Cora dataset is a citation network consisting of 2,708 nodes (papers) and 10,556 edges (citations). The Texas dataset represents a webpage graph with 183 nodes (web pages) and 499 edges (hyperlinks). Lastly, the grid graph is a two-dimensional $10 \times 10$ regular grid with 4-neighbor connectivity. For all three graphs, node features are randomly initialized from a normal distribution with a mean of 0 and variance of 1. These node features are then propagated over 80 layers (or iterations) using untrained GNNs to observe the energy dynamics.

Table 3: The grid of hyperparameters employed during model selection for the graph property prediction tasks (*GraphProp*), and `peptides-func` and `peptides-struct`.

| Hyperparameters | Values | |
| --- | --- | --- |
| | *GraphProp* | `peptides- (func, struct)` |
| Optimizer | Adam | AdamW |
| Learning rate | 0.003 | 0.001 |
| Weight decay | $10^{-6}$ | - |
| N. Layers | 10 | 40,17 |
| embedding dim | 20, 30 | 105 |
| $\sigma$ | tanh | ReLU |
| eig($\Lambda$) | 0.5, 0.75, 1.0 | 1.0 |

**Graph Property Prediction.** This experiment consists of predicting two node-level (i.e., eccentricity and single source shortest path) and one graph-level (i.e., graph diameter) properties on synthetic graphs sampled from different distribution, i.e., Erdős–Rényi, Barabasi-Albert, grid, caveman, tree, ladder, line, star, caterpillar, and lobster. Each graph contains between 25 and 35 nodes, with nodes assigned with input features sampled from a uniform distribution in the interval $[0, 1)$. The target values correspond to the predicted graph property. The dataset consists of 5,120 graphs for training, 640 for validation and 1,280 for testing.

We employ the same experimental setting and data outlined in [47]. Each model is designed as three components: the encoder, the graph convolution, and the readout. We perform hyperparameter tuning via grid search, optimizing the Mean Square Error (MSE). The models are trained using the Adam optimizer for a maximum of 1500 epochs, with early stopping based on the validation error, applying a 100 epochs patience. For each model configuration, we perform 4 training runs with different weight initializations and report the average results. We report in Table 3 the employed grid of hyperparameters.

**Long-Range Graph Benchmark.** We consider the `peptides-func` and `peptides-struct` datasets from [36]. Both datasets consist of 15,535 graphs, where each graph corresponds to 1D amino acid chain (i.e., peptide), where nodes are the heavy atoms of the peptide and edges are the bonds between them. `peptides-func` is a multi-label graph classification dataset whose objective is to predict the peptide function, such as antibacterial and antiviral function. `peptides-struct` is a multi-label graph regression dataset focused on predicting the 3D structural properties of peptides, such as the inertia of the molecule and maximum atom-pair distance.

We use the same experimental setting and splits from [36]. We perform hyperparameter tuning via grid search, optimizing the Average Precision (AP) in the Peptides-func and Mean Absolute Error (MAE) in the Peptide-struct. The models are trained using the AdamW optimizer for a maximum of 300 epochs. For each model configuration, we perform four training runs with different weight initializations and report the average results. We report in Table 3 the employed grid of hyperparameters.

**Tested Hyperparameters.** In Table 3 we report the grid of hyperparameters employed in our experiments by our method.

All experiments were run on a single NVIDIA RTX4090 GPU.

# D   Additional empirical results

In this section, we propose additional empirical results on over-smoothing and over-squashing, as well as the eigendistribution of the layerwise Jacobians of various standard GNNs.

## D.1   Additional Over-Smoothing Results

Here, we include additional results related to over-smoothing experiments. Figure 6 shows the effect of $||\Lambda||_2$ in GCN-SSM on different graph structures (similarly Figure 7 for GAT-SSM), showing that

lower Jacobian norms leads to a rapid decay of the Dirichlet energy, whereas values closer to one result in a more stable energy evolution. This result is also confirmed by Figure 8 and Figure 9. The former presents the vectorized Jacobian for ADGN [47], SWAN [48], and PHDGN [59] on Cora, while the latter the Dirichlet energy evolution of different models on different topologies. Notably, in Figure 9, ADGN, SWAN, and PHDGN exhibit stable Dirichlet energy across layers, and Figure 8 reveals that these Jacobian norms are close to one. These results confirm that stable dynamics also ensure a non-decaying Dirichlet energy, effectively preventing over-smoothing.

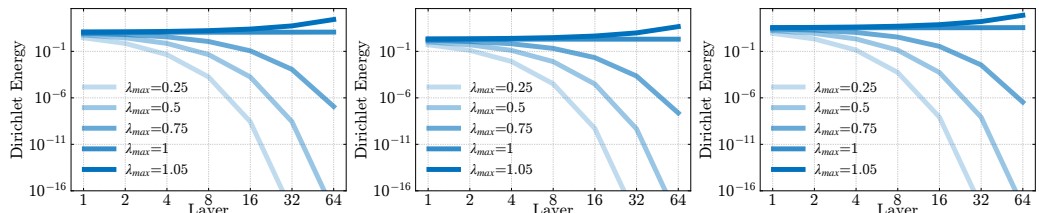

Figure 6: Dirichlet Energy evolution of GCN-SSM for different $||\Lambda||_2$ on different graph topologies. **Left:** Cora. **Middle:** Grid graph. **Right:** Texas.

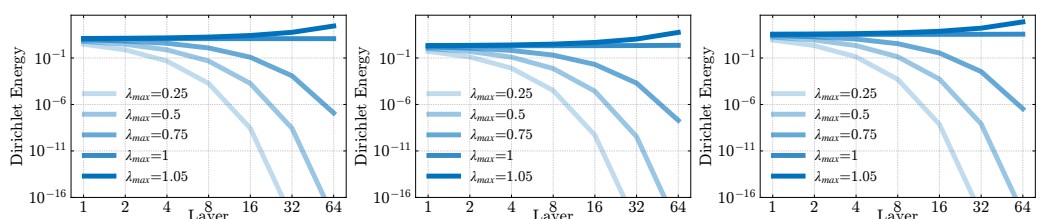

Figure 7: Dirichlet Energy evolution of GAT-SSM for different $||\Lambda||_2$ on different graph topologies. **Left:** Cora. **Middle:** Grid graph. **Right:** Texas.

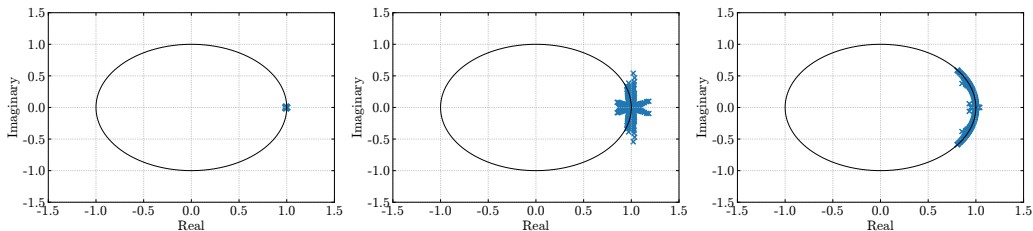

Figure 8: Vectorized Jacobian for ADGN [47], SWAN [48], and PHDGN [59] on Cora. **Left:** ADGN. **Middle:** SWAN. **Right:** PHDGN.

## D.2 Link between delay and vanishing gradients

Here, we show how the delay term in [55] is directly related to preventing vanishing gradients. We do so by showing that adding the delay term to a GCN is effective at preventing over-smoothing, see Figure 10, as well as by checking the histogram of eigenvalues of the Jacobian, see Figure 12. Figure 11 shows the effect of $||\Lambda||_2$ in DRew-SSM on the performance on the RingTransfer task, showing that lower Jacobian norms leads to a rapid performance decay, i.e., poor long-range propagation.

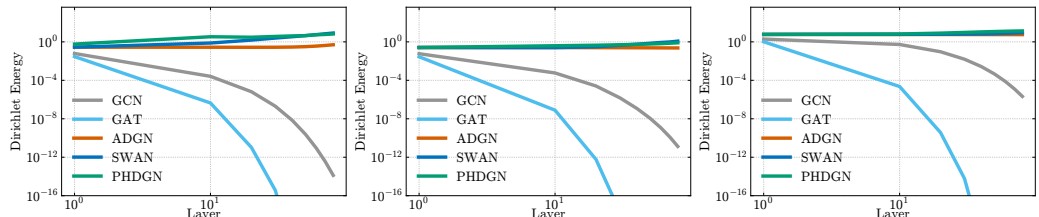

Figure 9: Dirichlet Energy evolution of different models on different topologies. **Left:** Cora. **Middle:** Grid graph. **Right:** Texas.

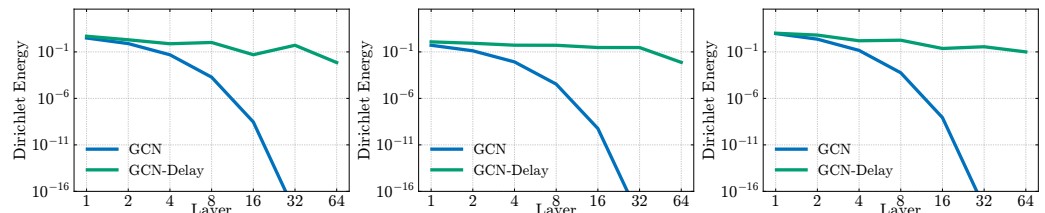

Figure 10: Dirichlet Energy evolution of GCN (+delay mechanism) on different topologies. **Left:** Cora. **Middle:** Grid graph. **Right:** Texas.

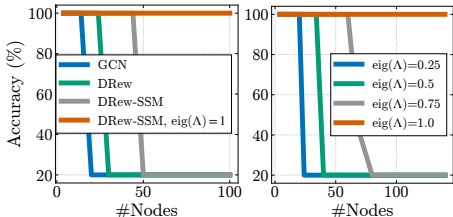

Figure 11: **Left:** Performance on the RingTransfer task for DRew [55]. **Right:** Effect of dissipativity on performance.

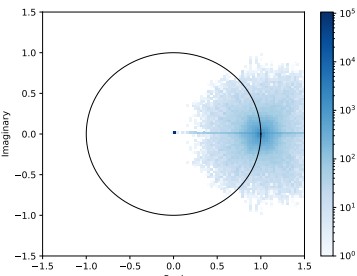

Figure 12: Eigenvalue distribution of DRew-GCN+delay on the ring transfer task.

## D.3 Graph Property Prediction

**Edge-of-chaos behavior and long-range propagation.** To further support our claim that mitigating gradient vanishing is key to strong long-range performance, Figure 13 shows each method's average Jacobian eigenvalue distance to the edge-of-chaos (EoC) region. The figure demonstrates that methods such as ADGN [47] and SWAN [48], which remain closer to EoC, effectively propagate information over large graph radii, resulting in superior performance across all three tasks. Figure 14 presents an ablation study on multiple ADGN variants, controlled by the hyperparameter $\gamma$, which governs the positioning of the Jacobian eigenvalues ($\gamma < 0$ places them outside the stability region, $\gamma > 0$ inside, and $\gamma = 0$ on the unit circle). Notably, regardless of the initial value of $\gamma$, ADGN consistently converges towards the EoC region as performance improves.

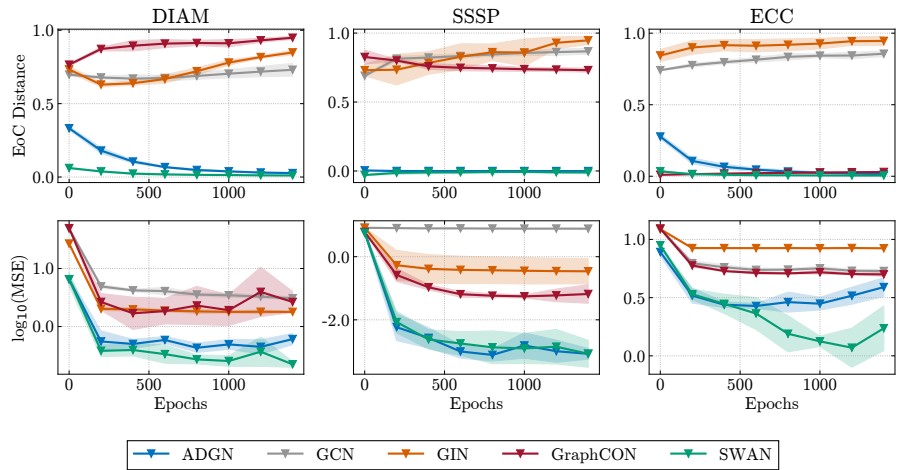

Figure 13: Performance on Graph Property Prediction tasks and average Jacobian eigenvalue distance to the edge of chaos (EoC) region for different GNN models.

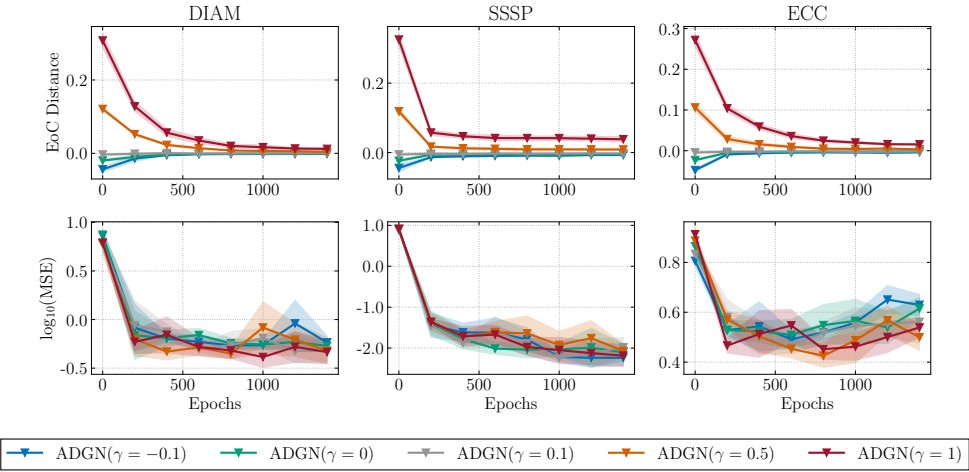

Figure 14: Performance on Graph Property Prediction tasks and average Jacobian eigenvalue distance to the edge of chaos (EoC) region for different ADGN dynamics, i.e., $\gamma \in [-0.1, 1]$. Negative values of $\gamma$ places the eigenvalues of the ADGN Jacobian outside the stability region, otherwise for positive values.

**Complete results.** Table 4 compares our method on graph property prediction tasks against a range of state-of-the-art approaches, including GCN [67], GAT [101], GraphSAGE [56], GIN [106], GC-NII [26], DGC [103], GRAND [23], GraphCON [91], ADGN [47], SWAN [48], PH-DGN [59], and DRew [55]. Our method achieves exceptional results across all three tasks, consistently surpassing MPNN baselines, differential equation-inspired GNNs, and multi-hop GNNs. These findings underscore how combining powerful model dynamics with improved connectivity provides substantial benefits in tasks that require long-range information propagation.

Table 4: Mean test set $log_{10}(\text{MSE})(\downarrow)$ and std averaged on 4 random weight initializations on Graph Property Prediction tasks. The lower, the better. Baseline results are reported from [47, 48, 59].

| Model | Diameter | SSSP | Eccentricity |
|---|---|---|---|
| **MPNNs** | | | |
| GCN | $0.7424_{\pm 0.0466}$ | $0.9499_{\pm 0.0001}$ | $0.8468_{\pm 0.0028}$ |
| GAT | $0.8221_{\pm 0.0752}$ | $0.6951_{\pm 0.1499}$ | $0.7909_{\pm 0.0222}$ |
| GraphSAGE | $0.8645_{\pm 0.0401}$ | $0.2863_{\pm 0.1843}$ | $0.7863_{\pm 0.0207}$ |
| GIN | $0.6131_{\pm 0.0990}$ | $-0.5408_{\pm 0.4193}$ | $0.9504_{\pm 0.0007}$ |
| GCNII | $0.5287_{\pm 0.0570}$ | $-1.1329_{\pm 0.0135}$ | $0.7640_{\pm 0.0355}$ |
| **Differential Equation inspired GNNs** | | | |
| DGC | $0.6028_{\pm 0.0050}$ | $-0.1483_{\pm 0.0231}$ | $0.8261_{\pm 0.0032}$ |
| GRAND | $0.6715_{\pm 0.0490}$ | $-0.0942_{\pm 0.3897}$ | $0.6602_{\pm 0.1393}$ |
| GraphCON | $0.0964_{\pm 0.0620}$ | $-1.3836_{\pm 0.0092}$ | $0.6833_{\pm 0.0074}$ |
| ADGN | $-0.5188_{\pm 0.1812}$ | $-3.2417_{\pm 0.0751}$ | $0.4296_{\pm 0.1003}$ |
| SWAN | $-0.5981_{\pm 0.1145}$ | $-3.5425_{\pm 0.0830}$ | $-0.0739_{\pm 0.2190}$ |
| PH-DGN | $-0.5473_{\pm 0.1074}$ | $\mathbf{-4.2993}_{\pm 0.0721}$ | $-0.9348_{\pm 0.2097}$ |
| **Graph Transformers** | | | |
| GPS | $-0.5121_{\pm 0.0426}$ | $-3.5990_{\pm 0.1949}$ | $0.6077_{\pm 0.0282}$ |
| **Multi-hop GNNs** | | | |
| DRew-GCN | $-2.3692_{\pm 0.1054}$ | $-1.5905_{\pm 0.0034}$ | $-2.1004_{\pm 0.0256}$ |
| + delay | $-2.4018_{\pm 0.1097}$ | $-1.6023_{\pm 0.0078}$ | $-2.0291_{\pm 0.0240}$ |
| **Our** | | | |
| GCN-SSM | $-2.4312_{\pm 0.0329}$ | $-2.8206_{\pm 0.5654}$ | $-2.2446_{\pm 0.0027}$ |
| + eig($\Lambda$) $\approx 1$ | $\underline{-2.4442}_{\pm 0.0984}$ | $-3.5928_{\pm 0.1026}$ | $\underline{-2.2583}_{\pm 0.0085}$ |
| + k-hop | $\mathbf{-3.0748}_{\pm 0.0545}$ | $\underline{-3.6044}_{\pm 0.0291}$ | $\mathbf{-4.2652}_{\pm 0.1776}$ |

## D.4 Additional comments on LRGB tasks

In our experiments with the LRGB tasks, we observe that the `peptides-func` task exhibits significantly longer-range dependencies than the `peptides-struct` task. Notably, the `peptides-struct` task performs best when the model is not initialized at the edge of chaos and requires fewer layers. Conversely, on `peptides-func` the model performs best when it is set to be at the edge of chaos, and shows a monotonic performance increase with additional layers, with optimal results achieved when using forty layers.

Furthermore, we highlight that while our experiments with a small hidden dimension adhere to the parameter budget established in [36], increasing the hidden dimension ($d \uparrow$) to 256 causes us to exceed the 500k parameter budget limit, even though our model maintains the same number of parameters as a regular GCN. While this budget is a useful tool to benchmark different models, we highlight that this restriction results in models running with fewer layers and small hidden dimensions. However, a large number of layers is crucial for effective long-range learning in graphs that are not highly connected, while increasing the hidden dimension also directly affects the bound in Theorem 5.1. As such, we believe that this parameter budget indirectly benefits models with higher connectivity graphs, inadvertently hindering models that do not perform edge addition.

To further strengthen the comparison on real-world tasks, in Table 5 we report results against GRED [32], a method inspired by state-space models (SSMs). We note that our model achieves superior performance on peptides-func, while GRED performs better on peptide-struct. Although GRED shares a few conceptual similarities with our approach, we emphasize several key differences: (i) GRED aggregates information from multiple neighborhoods at each layer, whereas our model aggregates from a single-hop neighborhood per layer. (ii) GRED's SSM update operates inward, beginning with distant nodes and moving toward the root node, whereas in our model each additional layer aggregates from progressively more distant neighbors, moving outward. This propagation pattern is more consistent with the standard MPNN framework. (iii) Our model employs a fixed and non-learned state matrix, removing the need for additional constraints to guarantee stable learning dynamics. (iv) Unlike GRED, we do not perform weight sharing when applying k-hop aggregation. Finally, we stress that the primary goal of our model design is not to achieve state-of-

the-art performance (as is the case with GRED) but rather to construct a minimal and controllable framework that allows us to isolate and study specific phenomena of interest.

Table 5: Comparison with GRED on the LRGB dataset. $d \uparrow$ adds latent dims.

| Model | Pept-func AP↑ | Pept-struct MAE↓ $(\times 10^{-2})$ |
|---|---|---|
| kGCN-SSM | $69.02_{+0.22}$ | $28.98_{+0.32}$ |
| $+d \uparrow$ | $\mathbf{72.12}_{+0.27}$ | $27.01_{+0.07}$ |
| GRED | $70.85_{+0.27}$ | $\mathbf{25.03}_{+0.19}$ |

## D.5 Scalability Results

In terms of runtime, GNN-SSM was deliberately designed to retain the simplicity and efficiency of its backbone (e.g., GCN). Our formulation introduces only two additional (fixed) matrices to store w.r.t. its backbone, and involves one element-wise addition and two extra matrix multiplications per layer. These additions have a negligible impact on memory and runtime. Consequently, our model retains the complexity of its backbone while substantially improving performance, see Tables 6 and 7 below.

Table 6: Epoch Time (sec.) for GCN and GCN-SSM when performing node classification of the Cora dataset.

| Layers | GCN | GCN-SSM |
|---|---|---|
| 5 | 0.009 | 0.009 |
| 10 | 0.015 | 0.017 |
| 20 | 0.025 | 0.031 |
| 30 | 0.041 | 0.046 |
| 40 | 0.053 | 0.051 |
| 50 | 0.066 | 0.075 |
| 60 | 0.078 | 0.089 |

Table 7: Accuracy on `ogbn-arxiv`[62].

| Model | ogbn-arxiv |
|---|---|
| **MPNNs** | |
| GAT | $72.01_{+0.20}$ |
| GCN | $70.84_{+0.23}$ |
| GraphSAGE | $71.49_{+0.27}$ |
| **Graph Transformers** | |
| NodeFormer | $59.90_{+0.42}$ |
| GraphGPS | $70.92_{+0.04}$ |
| GOAT | $72.41_{+0.40}$ |
| EXPHORMER + GCN | $72.44_{+0.28}$ |
| SPEXPHORMER | $70.82_{+0.24}$ |
| **Our** | |
| GCN-SSM | $\mathbf{72.49}_{+0.16}$ |

## D.6 State-Space Matrices Sensitivity

Empirically, we have that sharing $\Lambda$ across layers did not alter performance: using a single fixed $\Lambda$ yielded the same accuracy as training each $\Lambda_i$ with identical dynamics. Fixing $\Lambda$ ensures the system remains at the edge of chaos during training. Preserving this prior under a trainable $\Lambda$ would require additional constraints, in line with stabilization techniques used in RNN architectures, see Table 8 below.

Table 8: Performance comparison for different design choices when performing node classification of the Cora dataset.

| Layers | GCN-SSM (shared) | GCN-SSM (no sharing) | GCN-SSM (trained $\Lambda$) |
|--------|-----------------|---------------------|----------------------------|
| 5 | 76.3 | 78.3 | 71.3 |
| 10 | 78.5 | 78.5 | 71.1 |
| 20 | 81.2 | 77.8 | 49.9 |
| 30 | 78.1 | 79.6 | 33.8 |
| 40 | 77.5 | 78.5 | 31.9 |
| 50 | 76.4 | 74.6 | 31.9 |
| 60 | 77.8 | 77.4 | 31.9 |

## D.7 Preliminary Results on Heterophilic benchmarks

To further complement the empirical evaluation of our method, we conducted experiments on three heterophilic tasks from [85]: Amazon Ratings, Roman Empire, and Minesweeper. We compared GCN-SSM against the original GCN results reported in [85], as well as a GCN with the same depth as our model (denoted GCN (Optimal L)). The depth for each task was tuned based on validation performance, resulting in optimal layer counts of 4 for Amazon Ratings, 9 for Roman Empire, and 12 for Minesweeper. Results, averaged over three random seeds, are reported in Table 9 and show that GCN-SSM outperforms the original GCN by approximately 8 accuracy points on average, and exceeds GCN (Optimal L) by 31 accuracy points.

Table 9: Performance comparison of our GCN-SSM with respect to the original GCN results reported in [85], as well as a GCN with the same depth as our model (denoted GCN (Optimal L)) on the heterophilic datasets from [85].

| Model | Amazon Ratings | Roman Empire | Minesweeper |
|-------|:--------------:|:------------:|:-----------:|
| | Acc↑ | Acc↑ | AUC↑ |
| GCN ([85]) | $47.70_{\pm 0.63}$ | $73.69_{\pm 0.74}$ | $89.75_{\pm 0.52}$ |
| GCN (Optimal L) | $47.57_{\pm 0.67}$ | $33.83_{\pm 0.41}$ | $60.84_{\pm 0.89}$ |
| GCN-SSM (Optimal L) | $\mathbf{51.72}_{\pm 0.33}$ | $\mathbf{88.37}_{\pm 0.60}$ | $\mathbf{96.02}_{\pm 0.52}$ |

# E    Additional Details and Comments on Over-Smoothing

## E.1    Choice of Feature Distance Measure

Throughout the paper we adopt the *unnormalized Dirichlet energy*

$$\mathcal{E}(\mathbf{H}) \;=\; \mathrm{tr}\left(\mathbf{H}^\top \mathbf{L}\, \mathbf{H}\right) \;=\; \sum_{(u,v)\in\mathcal{E}} \|\mathbf{h}_u - \mathbf{h}_v\|_2^2,$$

This choice aligns with several well cited papers in the over-smoothing literature [91, 90]. Moreover, the connection we unravel between vanishing gradients and over-smoothing also explains why techniques borrowed from recurrent architectures [91, 47] are expected to mitigate feature collapse in GNNs.

While our theoretical analysis focuses on $\mathcal{E}(\mathbf{H})$ for mathematical simplicity, we will now also evaluate an alternative smoothness measure to ensure our insights generalize beyond this choice. In particular, we use the smoothness measure employed in other over-smoothing works [105, 95]

$$\mu(\mathbf{H}) = \left\|\mathbf{H} - \mathbf{1}\,\gamma_{\mathbf{H}}\right\|_F, \quad \gamma_{\mathbf{H}} = \frac{\mathbf{1}^\top \mathbf{H}}{N},$$

We report in Figure 15 empirical experiments on this measure, which empirically shows that the qualitative trends predicted by our unnormalized-energy theory also manifest under this alternative metric. Although formal equivalence between these energies and our collapse proofs is not explored,

this empirical alignment provides strong justification for the broader applicability of our analysis to the broader literature on oversmoothing.

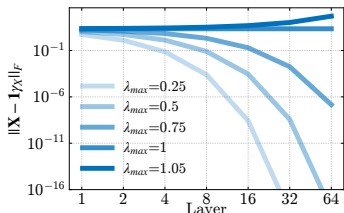

Figure 15: GCN-SSM on Cora with $\mu(\mathbf{H})$ smoothing measure.

## E.2 The effect of the Jacobian spectrum on node classification performance

In order to assess how the spectral properties of the layer-wise Jacobian influence node-classification performance, we carried out the following two experiments on the Cora citation network.

First, we systematically varied the spectrum of the diagonal $\Lambda$ matrix in our GCN-SSM backbone. For each number of layers $n \in \{5, 10, 20, 30, 40, 50, 60\}$, we set the maximal eigenvalue of $\Lambda$ to one of $\{1, 0.66, 0.33\}$, retrained the model, and recorded the best test accuracy. As shown in Table 10, moving the spectrum of $\Lambda$ away from unity leads to a pronounced degradation in accuracy, indicating that keeping the Jacobian eigenvalues near one is crucial for stable and effective information propagation across many layers.

Second, to isolate the contribution of the SSM backbone itself, we performed an analogous spectrum-shaping experiment directly on the weight matrix $\mathbf{W}$ of a vanilla GCN (i.e. without any SSM components). We scaled $\mathbf{W}$ so that its spectral radius lay near the edge of stability (spectral norm $\approx 1$), but otherwise left the model architecture and training unchanged. Despite matching the Jacobian stability regime, these "spectrally tuned" vanilla GCNs failed to achieve the accuracy improvements seen with the full GCN-SSM backbone (first column of Table 10). This confirms that merely tuning $\mathbf{W}$'s spectrum is insufficient: the structured state-space dynamics provided by the SSM backbone are essential for the observed performance gains.

Table 10: Node-classification accuracy on Cora when varying the spectrum of the backbone Jacobian ($\Lambda$) and comparing to vanilla GCN models whose weight matrix $\mathbf{W}$ is spectrally tuned near the edge of stability.

| $n_{\text{layers}}$ | $\text{eig}(\Lambda) = 1$ | $\text{eig}(\Lambda) = 0.66$ | $\text{eig}(\Lambda) = 0.33$ | $\text{eig}(W) = 1$ (GCN) |
|---|---|---|---|---|
| 5 | 81.30 | 78.10 | 74.00 | 71.90 |
| 10 | 78.70 | 61.90 | 56.00 | 33.80 |
| 20 | 78.90 | 48.00 | 30.20 | 31.80 |
| 30 | 80.00 | 39.70 | 18.00 | 22.30 |
| 40 | 77.90 | 34.60 | 20.30 | 25.60 |
| 50 | 77.70 | 29.10 | 24.90 | 23.80 |
| 60 | 77.70 | 20.50 | 20.40 | 29.10 |

## E.3 On Residual Connections and Gating

Several prior works have employed residual connections in GNNs to counteract over-smoothing, for example, see [69]. In fact, these residual-GNN designs can be viewed as a special case of our approach, corresponding to the choice $\mathbf{\Lambda} = \mathbf{I}$. Under this constraint, the model outperforms a standard, memoryless GCN (see Fig. 3), but only by accumulating node features in an unstructured way.

In order to guarantee stable propagation dynamics, the spectral properties of the propagation matrix $\mathbf{B}$ play an important role: by appropriately "damping" incoming signals, one can stabilize the system's behavior and prevent feature collapse or explosion.

To highlight the role of **B**, we conducted an ablation study on the Cora dataset in which the propagation matrix was removed from our method. Table 11 reports the results, showing that for deeper networks, the inclusion of **B** consistently improves performance. This behavior can be understood theoretically from the perspective of the Jacobian. While the eigenvalue structure induced by places the system near the edge of stability (with eigenvalues close to 1 in modulus), it is B that controls how much these dynamics "disperse" around the critical point (1,0) shown in Figure 2. In this setting, too much dispersion can lead to instability.

Table 11: Node-classification accuracy on Cora with and without the propagation matrix **B** across different number of layers.

| $n_{\text{layers}}$ | **Without B** | **With B** |
| --- | --- | --- |
| 2 | 80.2 | 79.6 |
| 4 | 77.8 | 80.4 |
| 8 | 80.0 | 79.8 |
| 16 | 74.1 | 79.8 |
| 32 | 31.9 | 79.9 |
| 64 | 31.9 | 79.9 |
| 128 | 14.9 | 77.9 |
| 256 | 13.0 | 80.2 |

## F  Supplementary Related Work and Limitations

**Long-range propagation and depth GNNs.**  Learning long-range dependencies on graphs involves effectively propagating and preserving information across distant nodes. Despite recent advancements, ensuring effective long-range communication between nodes remains an open problem [96]. Several techniques have been proposed to address this issue, including graph rewiring methods, such as [42, 100, 65, 8, 55, 13], which modify the graph topology to enhance connectivity and facilitate information flow. Similarly, Graph Transformers enhance the connectivity to capture both local and global interactions, as demonstrated by [109, 35, 97, 68, 87, 104]. Other approaches incorporate non-local dynamics by using a fractional power of the graph shift operator [74], leverage quantum diffusion kernels [73], regularize the model's weight space [47, 48, 49], exploit port-hamiltonian dynamics [59], or use a graph adaptive method based on a learnable ARMA framework [37]. Some methods which have increased the depth of GNNs include [69, 71]. Alternative methods combine the spatial and spectral perspective of graph propagation to enable long-range communication between nodes, either by using spatially and spectrally parametrized graph filters [43] or spectral filters based on Chebyshev polynomials [57]. Recent work has also proposed new datasets to test long range dependencies [70], as well as ways to theoretically measure them [4].

Despite the effectiveness of these methods in learning long-range dependencies on graphs , they primarily introduce solutions to mitigate the problem rather than establishing a unified theoretical framework that defines its underlying cause.

**Vanishing gradients in sequence modelling and deep learning.**  One of the primary challenges in training recurrent neural networks lies in the vanishing (and sometimes exploding) gradient problem, which can hinder the model's ability to learn and retain information over long sequences. In response, researchers have proposed numerous architectures aimed at preserving or enhancing gradients through time. Examples include Unitary RNNs [2], Orthogonal RNNs [60], Linear Recurrent Units [81], and Structured State Space Models [53, 52]. By leveraging properties such as orthogonality, carefully designed parameterizations, or alternative update mechanisms, these models seek to alleviate gradient decay and capture longer-range temporal relationships more effectively. We highlight recent work that also extends these insights to transformer based architectures [6].

**Dynamical systems and physics inspired neural networks.**  Since the introduction of Neural ODEs in [27], there have been various methods that employ ideas of dynamical systems within neural networks, including continuous-time methods [89, 78, 19, 20, 12, 77, 21] or state-space approaches [25, 33, 34, 37]. Within graph neural networks, we highlight PDE-GCN [38], GRAND [23], BLEND [22] and Neural Sheaf Diffusion [14] in the static graph domain, while CTAN [50] and TG-ODE [51]

in the temporal graph domain [46]. Other approaches which leverage other type of physics-inspired inductive biases such as topological latent space modelling include [24, 54, 15, 92].

**Broader Impact, Limitations and Future Work.**  We believe our work opens up a number of interesting directions that aim to bridge the gap between graph and sequence modeling. In particular, we hope that this work will encourage researchers to adapt vanishing gradient mitigation methods from the sequence modeling community to GNNs, and conversely explore how graph learning ideas can be brought to recurrent models. In our work, we mostly focused on GCN and GAT type updates, but we believe that our analysis can be extended to understand how different choices of updates and non-linearities affect training dynamics, which we leave for future work.

