# OpenReview forum: "On Vanishing Gradients, Over-Smoothing, and Over-Squashing in GNNs: Bridging Recurrent and Graph Learning"
_NeurIPS.cc/2025/Conference — NeurIPS 2025 poster_

### Official Review · Reviewer_VdM3 · 2025-06-30

**Clarity:** 4
**Significance:** 3
**Originality:** 3
**Rating:** 5
**Confidence:** 5

**Summary:**

This paper investigates the problem of vanishing gradients in Message-Passing Neural Networks (MPNNs) through a theoretical analysis grounded in signal processing. The authors highlight that many MPNNs experience what they term "extreme gradient vanishing", a phenomenon closely tied to over-smoothing. To address this, they introduce a state-space formulation for GNNs, referred to as kGNN-SSM, and demonstrate that in certain settings, this model avoids vanishing gradients and performs well on some long-range tasks.

**Questions:**

- Please see the "Weaknesses" section above and the conclusions.

- It is unclear to me if any kind of positional encodings were used. Mentioning this in Table 3 would be useful.

**Ethical Concerns:**

["NO or VERY MINOR ethics concerns only"]

**Final Justification:**

- The authors have performed new experiments on the heterophilic datasets from [Platonov2023] and on ogbn-arxiv, demonstrating that their proposed model has good results on node-level tasks.

- The authors have further analyzed the role of the input matrix B.

- The authors had satisfying responses to the other reviewer's comments.

---

The authors have briefly discussed the behaviour of their model when gating mechanisms are included, and have concluded that they ``interfered with training dynamics in non-trivial ways''. While I do think that a discussion regarding gating is a great addition, I do not think it is necessary for the paper to be accepted, and I acknowledge the limited rebuttal time.

Overall, I believe that the paper is solid and it should be accepted.

**Limitations:**

The authors should discuss their limited empirical evaluation for the kGNN-SSM model if additional experiments will not be performed.

**Quality:**

3

**Strengths And Weaknesses:**

**Strengths**:

1. The paper is very well-written.
2. While prior work has suggested a connection between vanishing gradients and issues like over-squashing and over-smoothing, this paper does an excellent job of formalizing those intuitions. The authors present a rigorous theoretical framework that makes these relationships explicit and clarifies a set of long-standing assumptions in the community.
3. The paper also effectively links over-smoothing to the limited long-range capabilities of MPNNs, adding further clarity to how these phenomena affect their performance.
4. The authors offer several insightful observations on why standard approaches for handling long-range dependencies or representation collapse, especially those borrowed from sequence modeling, may not translate effectively to GNNs. For instance, the comment from lines 164–165 offers a particularly clear example.
5.  I find the empirical analysis of the different GNN models to be insightful (e.g., Figs. 2 - 3, Figs. 5 - 13).

**Weaknesses**:

1. Minor typo: line 108: GNNs are known.
2. Minor typo: I believe that the authors wanted to refer to peptides-func on Line 823 - "Conversely, on peptides-func (...)".
3. The font for some figures should be increased. Figure 2 is particularly hard to read.
4. The authors might consider adding a related work discussion regarding the connection between their analysis and recently proposed spectral methods for modelling long-range interactions, such as [[4]] or [[5]].
5. While the theoretical analysis is strong and the authors connect vanishing gradients to over-smoothing and over-squashing in a compelling way, the proposed GNN-SSM model is very similar to prior work, such as GRED [[1]], with the main difference being the use of a custom SSM formulation instead of an LRU. As noted by the authors in Line 215, one could easily incorporate mechanisms like gating or other constraints on the state transition matrix, but the paper does not explore such variants. This makes the model feel more like a proof of concept rather than a technically novel contribution. The paper misses an opportunity to explore these minimal architectural variations of the GNN-SSM. Even a small set of experiments testing whether design choices like gating improve performance would have helped assess the proposed approach better.
6. The empirical evaluation of GNN-SSM is limited in both scope and depth. The model is only tested on the Peptides datasets from the LRGB, while other relevant datasets like PCQM-Contact are excluded. Moreover, the analysis focuses mainly on long-range tasks and does not consider standard graph benchmarks such as QM9, ZINC, ogbn-arxiv, or ogbn-proteins. Including these would help determine whether GNN-SSM generalizes beyond specialized cases.
7. It is unclear to me whether the peptides-func dataset is a truly reliable indicator of long-range modeling capabilities. Prior work has shown that adding positional encodings to standard MPNNs (such as a GCN) can already lead to strong performance [[3]], which calls into question how much the gains on peptides-func are due to actual long-range modelling. A more targeted evaluation, such as the mechanistic in-context learning task from S2GNN [[4]], would better demonstrate the model’s ability to handle long-range dependencies in practice.
8. The evaluation of over-smoothing effects would be more convincing if it included better heterophilic datasets. Although the authors report results on the Texas dataset from the WebKB collection, this collection has been heavily scrutinized in recent work, and more reliable alternatives have been proposed [[2]]. I recommend including at least one of these newer datasets to provide a more solid quantitative analysis.

**Conclusions**:

Overall, I appreciate the theoretical framework developed by the authors. Formalizing the link between vanishing gradients, over-smoothing, and over-squashing adds valuable clarity to ongoing discussions around the limitations of MPNNs, while also building meaningful connections to the sequential modeling literature. The idea of permutation-equivariant GNN-SSMs is compelling, and the theoretical setup provides a strong justification for the model's design.

However, I found the empirical evaluation underwhelming. The proposed kGNN-SSM shares conceptual similarities with prior works such as Drew, GRED, and to some extent, with more recent models like S2GNN [[4]], which also draws parallels between long-range graph learning and state-space models. The current evaluation does little to differentiate the proposed model or explore its capabilities in depth. As it stands, the model functions more as a theoretical proof-of-concept than a technically novel or well-tested contribution. While this framing is valid due to the more theoretical nature of the work, it feels like a missed opportunity to more fully explore the empirical potential of the proposed model. I would have preferred either a more concise theoretical discussion paired with deeper empirical analysis or a more extended version of the paper, which would maybe be better suited as a journal submission.

Scoring this paper was difficult. It is very well-written, and the theoretical contributions are sound and interesting. The authors demonstrate that both vanishing gradients and over-smoothing affect the long-range capabilities of MPNNs, and that they are, in fact, connected. However, the GNN-SSM model lacks technical novelty, and its empirical evaluation is limited. Despite the comprehensive comparisons between various GNN models in the Supplementary, the evaluation of the proposed GNN-SSM itself feels disconnected from the strong theoretical motivation.

My initial score is a 4 (Borderline Accept). That said, I am genuinely inclined toward accepting the paper and will raise my score to a 5 (Accept) or 6 (Strong Accept) if the authors address some of my concerns. The most valuable and "low-hanging" additions, in my opinion, would be evaluating GNN-SSM on the heterophilic datasets proposed in [[2]] and on more standard graph or node property prediction tasks (see Weakness 5). These datasets are relatively small, and the experiments should be feasible to run during the rebuttal period. Please note that I will not lower my score if GNN-SSM underperforms on these datasets; my concern is about completeness, not performance.

I would also be excited to see variants of GNN-SSM that incorporate gating or related architectural changes, and I believe that running the mechanistic in-context learning experiment from S2GNN [[4]] would provide especially strong evidence of long-range modeling capabilities. While I understand these additions may be difficult to implement within the rebuttal timeline, I highly recommend that the authors consider them for an eventual camera-ready version. In my view, they would significantly strengthen the paper.


[1]: https://arxiv.org/pdf/2312.01538v1
[2]: https://arxiv.org/pdf/2312.01538v1
[3]: https://arxiv.org/abs/2309.00367
[4]: https://arxiv.org/pdf/2405.19121
[5]: https://arxiv.org/pdf/2506.07624

---

> ### Author Rebuttal · Authors · 2025-07-30
>
> Thank you for your careful and detailed reviews. We're especially grateful that you highlighted some of the good qualities of our work and provided thoughtful suggestions on how to improve. Also, thank you for emphasizing the observation on lines 164–165, as this is a very subtle and important point, and we're glad it resonated with you. We hope that the revisions we've made in response to your comments have improved the paper and addressed your concerns.
>
> >Minor typo: line 108: GNNs are known.
> Minor typo: I believe that the authors wanted to refer to peptides-func on Line 823 - "Conversely, on peptides-func (...)".
> The font for some figures should be increased. Figure 2 is particularly hard to read.
>
> Thank you for pointing out these minor issues. We’ve corrected the typos at lines 108 and 823, and we’ve increased the font size in several figures, particularly in Figure 2, which we agree was difficult to read in its original form.
>
> >The authors might consider adding a related work discussion regarding the connection between their analysis and recently proposed spectral methods for modelling long-range interactions, such as [4] or [5].
>
> Thank you for this valuable suggestion. We find [4] particularly interesting for its elegant combination of spatial and spectral perspectives, and we agree that exploring potential connections with our recurrent mechanism would be a worthwhile direction for future work. As for [5], it indeed builds on ideas closely related to ours regarding signal propagation, and we will definitely cite it in the revised version.
>
> >While the theoretical analysis is strong and the authors connect vanishing gradients to over-smoothing and over-squashing in a compelling way, the proposed GNN-SSM model is very similar to prior work, such as GRED [1], with the main difference being the use of a custom SSM formulation instead of an LRU. As noted by the authors in Line 215, one could easily incorporate mechanisms like gating or other constraints on the state transition matrix, but the paper does not explore such variants. This makes the model feel more like a proof of concept rather than a technically novel contribution. The paper misses an opportunity to explore these minimal architectural variations of the GNN-SSM. Even a small set of experiments testing whether design choices like gating improve performance would have helped assess the proposed approach better.
>
> Thank you for this thoughtful and constructive comment. We have outlined the main conceptual and architectural differences between GNN-SSM and GRED in our response to Reviewer v1TX, particularly regarding our use of a custom SSM formulation with an emphasis on minimalism and interpretability.
>
> That said, we found your suggestion regarding gating mechanisms particularly relevant. We deliberately avoided incorporating more complex design elements (such as gating or structured transition matrices) in the initial version of our model, as our aim was to isolate and highlight the role of vanishing gradients through targeted ablations. However, we agree that our manuscript could have explored the impact of such variations more explicitly.
>
> One component we believe plays a critical role in gating is the input matrix B, which modulates the influence of input signals on the recurrent dynamics. To initially test the influence of this, we performed an ablation study on the Cora dataset, in which we removed the B matrix. The results are shown below:
>
> | # Layers (n) | No **B** | **B** |
> |--------------|----------|-------------------------------|
> | 2            | 80.2     | 79.6                          |
> | 4            | 77.8     | 80.4                          |
> | 8            | 80.0     | 79.8                          |
> | 16           | 74.1     | 79.8                          |
> | 32           | 31.9     | 79.9                          |
> | 64           | 31.9     | 79.9                          |
> | 128          | 14.9     | 77.9                          |
> | 256          | 13.0     | 80.2                          |
>
>
> This behavior can be understood theoretically from the perspective of the Jacobian: While the eigenvalue structure induced by $\Lambda$ places the system near the edge of stability (with eigenvalues close to 1 in modulus), it is B that controls how much these dynamics "disperse" around the critical point (1,0) shown in Figure 2. In this setting, too much dispersion can lead to instability.
>
> To test this further, we scaled the B matrix by a constant $\alpha=$(1/nlayers), which should, in principle, make the eigenvalues closer to the critical point by reducing dispersion while still enabling information to be processed by the network. We found that this scaled variant significantly stabilizes the loss and accelerates convergence, requiring roughly an order of magnitude fewer epochs to reach a solution on the Cora dataset. Although we are unable to include additional plots in the rebuttal, we will present these findings in the camera-ready version in a dedicated section, as this appears to be a promising direction for future research.
>
> We also experimented with a variety of other gating mechanisms during the rebuttal, but found that many interfered with training dynamics in non-trivial ways. Nevertheless, we think that the consistent benefits we observed from appropriately modulating B underscore the importance of exploring such mechanisms, and we will likely investigate this in future work.
>
> Thank you again for the insightful suggestion.
>
> > The most valuable and "low-hanging" additions, in my opinion, would be evaluating GNN-SSM on the heterophilic datasets proposed in [2] and on more standard graph or node property prediction tasks (see Weakness 5).
>
> As we noted before, our goal was not to develop a high-performing model, but rather to design a variant that is deliberately minimal and controllable. This minimalism allows us to cleanly isolate and analyze the role of effective rewiring and stable gradient propagation, two mechanisms we argue are central to over-smoothing and over-squashing. That said, we fully understand your concern regarding empirical completeness, and we’ve taken it seriously in our rebuttal.
>
> Regarding the heterophilic datasets, we believe the link in reference [2] might have been a typo, as it pointed us to the GRED paper. However, we assume you were referring to the more recent benchmark introduced in [Platonov2023]. In response, we evaluated GCN-SSM on three of the datasets from that benchmark, comparing it against (i) the original GCN results reported in the paper, (ii) GCN with the same depth as our model (denoted "GCN (Optimal L)"), and (iii) our GCN-SSM with the same depth. Since GCN-SSM benefits from a larger number of layers, we tuned the depth for each task based on performance on the validation set. Results are shown below (for three random seeds):
>
> | Dataset         | GCN (Orig)     | GCN (Optimal L) | GCN-SSM (Optimal L) |
> |------------------|----------------|------------------|----------------------|
> | Amazon Ratings    | 47.7 ± 0.63    | 47.57  ± 0.67         | 51.72  ± 0.33            |
> | Roman Empire      | 73.69 ± 0.74   | 33.83   ± 0.41         | 88.37   ± 0.60               |
> | Minesweeper       | 89.75 ± 0.52   | 60.84     ± 0.89        | 96.02     ± 0.52            |
>
> We used the following number of layers for each dataset:
> | Dataset         | # Layers (n) |
> |------------------|---------------|
> | Amazon Ratings    | 4             |
> | Roman Empire      | 9             |
> | Minesweeper       | 12            |
>
> We did not run full-scale experiments on all datasets in the collection, as tuning the number of layers requires multiple training runs, and we were constrained by the rebuttal timeline. Nonetheless, we believe these results already offer meaningful evidence of GCN-SSM’s generalization to challenging heterophilic tasks.
>
> We also evaluated GCN-SSM on some ogbn-arxiv, as suggested by the reviewer:
>
> | Dataset     | GCN             | GCN-SSM         |
> |--------------|------------------|------------------|
> | ogbn-arxiv   | 70.84 ± 0.23     | 72.49 ± 0.16     |
>
> Due to time constraints (e.g., hyperparameter tuning, optimal depth selection), we were only able to include this limited set of new experiments in the rebuttal. However, we plan to significantly expand our empirical evaluation in the camera-ready version. This will include further experiments on the heterophilic benchmarks from [Platonov2023], more standard graph-level datasets, and the mechanistic in-context learning experiment from S2GNN [4]—which we agree would provide compelling evidence of long-range modeling ability.
>
> Once again, we greatly appreciate your thoughtful feedback. Your comments have had a clear and direct impact on the direction of this work, and we are hopeful that the revisions and additional experiments presented here will merit a more favorable assessment.
>
> [Platonov2023] Platonov, Oleg, et al. "A critical look at the evaluation of GNNs under heterophily: Are we really making progress?." arXiv preprint arXiv:2302.11640 (2023).

---

> > ### Comment · Reviewer_VdM3 · 2025-08-05
> >
> > I appreciate the detailed rebuttal responses that the authors have made to me and the other reviewers. I now believe that the paper should be clearly accepted, given the expanded analysis and new experimental results on the heterophilic datasets from [Platonov2023] and ogbn-arxiv.
> >
> > > We also experimented with a variety of other gating mechanisms during the rebuttal, but found that many interfered with training dynamics in non-trivial ways. Nevertheless, we think that the consistent benefits we observed from appropriately modulating B underscore the importance of exploring such mechanisms, and we will likely investigate this in future work.
> >
> > I would highly encourage the authors to discuss these (preliminary) results separately in the Appendix. These observations (including negative results) might be highly valuable for future work or for practitioners.
> >
> > Once again, I thank the authors for their responses! I believe that the paper is in good shape with the new additions.

---

> > > ### Author Response · Authors · 2025-08-08
> > >
> > > We thank the reviewer for their positive feedback and engagement. We are glad that the reviewer believes our paper should be clearly accepted, given our detailed rebuttal responses and additional experiments. As suggested, we will include the discussion in the revised manuscript to aid future research.

---

### Official Review · Reviewer_ZDHY · 2025-07-01

**Clarity:** 4
**Significance:** 3
**Originality:** 3
**Rating:** 5
**Confidence:** 4

**Summary:**

This paper proposes to study problems associated with graph neural networks such as oversmoothing and oversquashing through the lens of vanishing gradients. Vanishing gradients were originally studied in recurrent neural networks, and this paper argues that vanishing gradients can also be used to study oversmoothing and oversquashing in GNNs. Gradients are already commonly used as a tool for quantifying oversquashing, but this paper uses vanishing gradients to unify the study of oversmoothing and oversquashing. The contributions of this paper are as follows:
- Theorem 3.2 proves that if the weight matrix is normally distributed with variance < 1, then the Jacobian of a GCN has eigenvalues < 1, so GCNs would experience vanishing gradients in this setting. Figures 2, 5, and 6 analyses the eigenvalues of real GNNs and show that Theorem 3.2 is true in practice.
- The paper proposes GNN-SSM as a remedy for the vanishing gradient problem posed by Theorem 3.2. GNN-SSM combined elements of classical GNNs with state-space models.
- Proposition 3.3 shows that the eigenvalues of the Jacobian of GNN-SSM are controlled both by a GNN update (which by Theorem 3.2 has small eigenvalues) and a state transition matrix. Thus, by controlling the eigenvalues of the state transition matrix, the eigenvalues of the entire Jacobian can be controlled. This is validated in Figure 2.
- The paper use vanishing gradient to prove why may GNNs experience oversmoothing. Proposition 4.4 shows that GNNs that experience vanishing gradients will also experience oversmoothing. It proves this by showing that the feature of GNNs with vanishing gradients will necessarilly converge to the zero vector as their norms shrink with each layer. This also implies that the Dirichlet energy of the node features will decrease after each layer. Figures 7 and 8 demonstrate that a GNN-SSM has constant Dirichlet energy across layers.
- The paper argue GNN-SSMs can also help alleviate oversquashing. It was previously argued that oversquashing was caused by small gradients between the features of different nodes. The authors argue that GNN-SSMs could increase the gradient between nodes, thereby alleviating oversquashing. One of the most common remedies for oversquashing is rewiring. Figure 4 shows that using rewiring in conjunction with GNN-SSMs improves over using either of them separately.
- The authors test GNN-SSMs on various real world datasets and show that GNN-SSMs can improve over vanilla GNNs in these settings. Tables 1, 2, and 4 show that GNN-SSMs typically outperform classical GNN baselines on these tasks.

**Questions:**

# Questions
- How are the state transition matrices $\Lambda$ computed, and how do you control their eigenvalues? Are these matrices learnable or fixed?
- In Figure 3, middle, what is model is this plotting the features for?

# Suggestions
- Figure 1 is a little hard to understand, even after reading the paper. What are the blue lines? Which circles correspond to features of the same node? What circle are at the same layer, and which are at different layers? I think this figure could be improved if each element was more clearly labelled.
- The boxes around the theorem statements look awkward as the text inside the box is too close to the edge. I would recommend using larger margins on the border of the box. Borders on the boxes and rounded corners may also improve the look of the boxes.

# Typos
Line 109: "over-smoothing [15, 68] describes the tendency". I am not sure what you intended to say here, but there must have been a typo as this sentence doesn't make sense.
Line 777: "were run in" -> "were run on"

**Ethical Concerns:**

["NO or VERY MINOR ethics concerns only"]

**Final Justification:**

After reading the rebuttal to my review, and the reviews and rebuttals of all other reviewers, I maintain my original opinion that this is a strong paper that is clearly worthy of acceptance at NeuRIPS.

**Limitations:**

Yes.

**Paper Formatting Concerns:**

None.

**Quality:**

4

**Strengths And Weaknesses:**

# Strengths

- The authors do an excellent job of validating their claims both theoretically and experimentally. Every theorem in the paper has an accompanying experiment showing this holds in practice and an experiment demonstrating how properly designed GNN-SSMs can alleviate the observed bad behavior. Thus, the authors demonstrate conclusively that GNNs face several problems that GNN-SSMs can help fix.
- The authors empirically show that GNN-SSMs can achieve strong performance on real datasets compared to many baselines.
- In this weaknesses, I mention that some of the theoretical results only prove sufficient but not necessary conditions for many of the claims in the paper. However, in these instances, the paper does a good job of augmenting the theoretical results with empirical evidence to show that fixing vanishing gradients can help fix the problem in question.

# Weaknesses

- Many of the theoretical results are one-sided. e.g., only proving a sufficient but not necessary condition for the expected behavior. For example, Proposition 3.3 only proves an upper bound on the Jacobian by the norm of the state transition matrix, so increasing the norm of the state transition matrix will not necessarily increase the norm of the Jacobian. Likewise, Proposition 4.4 show that vanishing gradients can lead to decreasing Dirichlet energy. Thus, Proposition 4.4 does not prove that decreasing Dirichlet energy only happens when there are vanishing gradients, and alleviating vanishing gradients is necessary but not necessarily sufficient to fix these oversmoothing.
Likewise, Theorem 3.2 only holds for GCNs, while ideally it would hold for more general message-passing networks like GINs or GATs.

---

> ### Author Rebuttal · Authors · 2025-07-30
>
> Thank you for the very positive and encouraging review. We're glad to hear that both the theoretical and empirical aspects of the paper came through clearly. We also appreciate your constructive comments on the scope of our theoretical results, and we hope that the clarifications and additions in our rebuttal help to address any remaining concerns.
>
> > Many of the theoretical results are one-sided. e.g., only proving a sufficient but not necessary condition for the expected behavior. For example, Proposition 3.3 only proves an upper bound on the Jacobian by the norm of the state transition matrix, so increasing the norm of the state transition matrix will not necessarily increase the norm of the Jacobian. Likewise, Proposition 4.4 show that vanishing gradients can lead to decreasing Dirichlet energy. Thus, Proposition 4.4 does not prove that decreasing Dirichlet energy only happens when there are vanishing gradients, and alleviating vanishing gradients is necessary but not necessarily sufficient to fix these oversmoothing. Likewise, Theorem 3.2 only holds for GCNs, while ideally it would hold for more general message-passing networks like GINs or GATs.
>
> Thank you for this thoughtful and technically precise feedback. As you rightly point out, many of our theoretical results establish sufficient (but not necessary) conditions. In some cases, such as Proposition 3.3, we only prove upper bounds (e.g., on the Jacobian norm via the state transition matrix). While this does not guarantee that increasing the norm of $\Lambda$ will necessarily increase the Jacobian norm, we observed empirically that this correspondence does hold in practice. We will make the limitations of these results clearer in the revised version.
>
> Your comment on oversmoothing is particularly insightful. Our core argument is that what is commonly referred to as "over-smoothing" largely stems from the contractive nature of typical GNNs, and that vanishing gradients are a primary underlying cause. While vanishing gradients are not the only mechanism that can lead to over-smoothing, we believe that it captures the majority of such failures observed in practice. In our view, the depth limitations in GNNs stem more from trainability issues than expressivity constraints due to convergence to rank-one representations.
>
> We would also like to clarify that while Theorem 3.2 focuses on GCNs, our analysis also extends to GATs, since attention-weighted adjacency matrices share the same contractive properties. As you correctly note, we do not currently formalize behavior for more general GNNs such as GIN. This is indeed more difficult to analyze rigorously, but one advantage of our Jacobian-spectrum-based analysis is that it can be applied to arbitrary GNN layers. Inspired by your suggestion and that of Reviewer v1TX, we computed the Jacobian spectrum for a randomly initialized GIN layer and found it to be highly unstable (with spectral norms around 150), resulting in divergent Dirichlet energy. This may help explain behaviors recently highlighted in [1], and we will include this analysis in the revised manuscript.
>
> >How are the state transition matrices $\Lambda$ computed, and how do you control their eigenvalues? Are these matrices learnable or fixed?
>
> Thank you for the question. To construct the state transition matrices $\Lambda$, we begin by generating a random unitary matrix, which has all eigenvalues on the unit circle. We then scale it to control the spectral radius. Alternatively, we can compute its eigendecomposition, manually place the eigenvalues where desired (e.g., near 1 in modulus), and reconstruct the matrix. This allows us to design $\Lambda$ with precise control over the system dynamics.
>
> In terms of training, we keep $\Lambda$ fixed. We made this decision to ensure controllable and stable dynamics, as we found that making $\Lambda$ learnable led to divergence and instability during training. We had mentioned this briefly in line 99, but based on your feedback, we will now include a more dedicated explanation in the revised manuscript.
>
> > In Figure 3, middle, what is model is this plotting the features for?
>
> Thank you for pointing that out. All plots in Figure 3 (including the middle one) are for the Cora dataset. We'll make this clearer in the revised version to avoid any confusion.
>
> > Figure 1 is a little hard to understand, even after reading the paper. What are the blue lines? Which circles correspond to features of the same node? What circle are at the same layer, and which are at different layers? I think this figure could be improved if each element was more clearly labelled
>
> Thank you for the feedback. We agree that Figure 1 can be clarified further. To address your questions:
> - The blue lines indicate how each node feature evolves across layers, that is, how the 2D feature vector for each node changes as more layers are added.
> - Each circle corresponds to a node’s 2D feature at a given layer and is placed in the x-y plane according to the feature values.
> - Circles connected by a blue line represent the same node across successive layers.
> - The color of each circle encodes the norm of the node feature.
>
> The goal of the figure is to illustrate that node features evolve through the network in a norm-preserving manner, without collapsing or contracting, which aligns with the theoretical results discussed in the paper. We will revise the caption to make these points clearer.
>
> > The boxes around the theorem statements look awkward as the text inside the box is too close to the edge. I would recommend using larger margins on the border of the box. Borders on the boxes and rounded corners may also improve the look of the boxes
> Line 109: "over-smoothing [15, 68] describes the tendency". I am not sure what you intended to say here, but there must have been a typo as this sentence doesn't make sense. Line 777: "were run in" -> "were run on"
>
> Thank you, we’ll fix these typos! We also appreciate the suggestion regarding the theorem boxes. We’ll increase the internal padding, and we’ll experiment with using borders and rounded corners to improve readability and visual consistency.
>
>
>
> [1] Arnaiz-Rodriguez, Adrian, and Federico Errica. "Oversmoothing," Oversquashing", Heterophily, Long-Range, and more: Demystifying Common Beliefs in Graph Machine Learning. arXiv preprint arXiv:2505.15547 (2025).

---

> > ### Comment · Reviewer_ZDHY · 2025-08-02
> >
> > Thank you for your reply. I appreciate the clarifications on all points.

---

### Official Review · Reviewer_CMgd · 2025-07-03

**Clarity:** 3
**Significance:** 2
**Originality:** 3
**Rating:** 4
**Confidence:** 3

**Summary:**

The paper presents a GNN as convolutional neural network with recurrent state updates. This connection is then exploited to create a state-space formulation of the GNN gradient descent updates. The nature of gradient update across the shared weights is shown as an evolution dynamics in state space. The problems of vanishing gradient and homogenization are explained in terms of this state-space dynamics. The issue of diminishing effect of distant nodes are also shown to exist.

Next, authors propose changes in the learning and update mechanism which arrest these problems. The state-space formulation allows a control theoretic analysis of the strategies and synthesis of effective update rules. The method is elegant.

**Questions:**

1. Does the proposed method scale to very large scale GNN where the challenges of vanishing gradient and homogenization are even more pronounced.

2. How does the proposed method compare with state of art GNN training techniques which are not based on state space formulations but use sophisticated optimizer and message passing heuristics.

3. Provide a discussion between the functional differences in the proposed method to avoid vanishing gradient in comparison with the known approaches in literature.

**Ethical Concerns:**

["NO or VERY MINOR ethics concerns only"]

**Limitations:**

The method is elegant. More results on large scale GNNs are required to consider the preent work as a state of art in GNN training.

Further experimental results comparing with state of art in GNN with respect to the identified challenges should be presented.

**Paper Formatting Concerns:**

No formatting concerns.

**Quality:**

3

**Strengths And Weaknesses:**

The strength of the paper is the elegant state-space formalization of the GNN learning problem. The effects of vanishing gradient and homogenization can be easily observed in this analysis framework. Mitigation of these challenges using control theoretic methods in the state-space is found to be effective. Experimental results demonstrate effectiveness of the proposed technique.

The theoretical formulation is nice. However, there are some concerns about practical applicability in large scale GNN. The resulting state-space may explode and many of the existing control theoretic methods my break-down in large state-spaces. The typical state-space sizes common in control theoretic problems are much smaller.

Comparison with state-of-art GNN training schemes are not presented in the experimental results section. particularly the ones which rely on modifications of the message passing techniques.

---

> ### Author Rebuttal · Authors · 2025-07-30
>
> Thank you for your constructive review. We respond to your questions and comments below.
>
> > Does the proposed method scale to very large scale GNN where the challenges of vanishing gradient and homogenization are even more pronounced.
> >How does the proposed method compare with state of art GNN training techniques which are not based on state space formulations but use sophisticated optimizer and message passing heuristics.
>
> Thank you for your question regarding scalability and comparison with more sophisticated GNN training strategies.
> In terms of computational complexity and scalability, GNN-SSM was deliberately designed to retain the simplicity and efficiency of its backbone (e.g., GCN). Our formulation introduces only two additional fixed matrices and involves one elementwise addition and two additional matrix multiplications per layer. This has a negligible effect on memory and runtime, particularly when compared to models with learnable global attention or deep message-passing schemes. As a result, GNN-SSM maintains the scalability of its base architecture, which we demonstrate below.
>
> | # Layers | GCN   | GCN-SSM |
> |----------|--------|-----------|
> | 5        | 0.009  | 0.009     |
> | 10       | 0.015  | 0.017     |
> | 20       | 0.025  | 0.031     |
> | 30       | 0.041  | 0.046     |
> | 40       | 0.053  | 0.051     |
> | 50       | 0.066  | 0.075     |
> | 60       | 0.078  | 0.089     |
>
> To further assess scalability and performance, we also evaluated GCN-SSM on the large-scale ogbn-arxiv benchmark.
> | Dataset     | GCN             | GCN-SSM         |
> |--------------|------------------|------------------|
> | ogbn-arxiv   | 70.84 ± 0.23     | 72.49 ± 0.16     |
>
>
> To contextualize this, we compare it with published results from a range of MPNN and Graph Transformer baselines, see [4]
>
> | Model              | ogbn-arxiv     |
> |--------------------|----------------|
> | GraphSAGE          | 71.49 ± 0.27   |
> | GAT                | 72.01 ± 0.20   |
> | NodeFormer         | 59.90 ± 0.42   |
> | GraphGPS           | 70.92 ± 0.04   |
> | GOAT               | 72.41 ± 0.40   |
> | EXPHORMER + GCN    | 72.44 ± 0.28   |
> | SPEXPHORMER        | 70.82 ± 0.24   |
>
> As the results show, GCN-SSM is competitive with or better than several recent high-performing models that rely on more complex architectures or training schemes, despite using a simpler, more interpretable design.
>
> Furthermore, the paper already includes extensive comparisons on synthetic graph algorithm benchmarks designed to test long-range reasoning capabilities, which includes state-of-the-art methods like DRew [5].  Below, we reproduce the relevant portion of the table for reference:
>
> | Model            | Diameter         | SSSP             | Eccentricity     |
> |------------------|------------------|------------------|------------------|
> | **MPNNs**        |                  |                  |                  |
> | GCN              | 0.7424 ± 0.0466  | 0.9499 ± 0.0001  | 0.8468 ± 0.0028  |
> | GAT              | 0.8221 ± 0.0752  | 0.6951 ± 0.1499  | 0.7909 ± 0.0222  |
> | GraphSAGE        | 0.8645 ± 0.0401  | 0.2863 ± 0.1843  | 0.7863 ± 0.0207  |
> | GIN              | 0.6131 ± 0.0990  | -0.5408 ± 0.4193 | 0.9504 ± 0.0007  |
> | GCNII            | 0.5287 ± 0.0570  | -1.1329 ± 0.0135 | 0.7640 ± 0.0355  |
> | **DiffEq-inspired GNNs** |         |                  |                  |
> | DGC              | 0.6028 ± 0.0050  | -0.1483 ± 0.0231 | 0.8261 ± 0.0032  |
> | GRAND            | 0.6715 ± 0.0490  | -0.0942 ± 0.3897 | 0.6602 ± 0.1393  |
> | GraphCON         | 0.0964 ± 0.0620  | -1.3836 ± 0.0092 | 0.6833 ± 0.0074  |
> | ADGN             | -0.5188 ± 0.1812 | -3.2417 ± 0.0751 | 0.4296 ± 0.1003  |
> | SWAN             | -0.5981 ± 0.1145 | -3.5425 ± 0.0830 | -0.0739 ± 0.2190 |
> | PH-DGN           | -0.5473 ± 0.1074 | -4.2993 ± 0.0721 | -0.9348 ± 0.2097 |
> | **Graph Transformers** |         |                  |                  |
> | GPS              | -0.5121 ± 0.0426 | -3.5990 ± 0.1949 | 0.6077 ± 0.0282  |
> | **Multi-hop GNNs** |               |                  |                  |
> | DRew-GCN         | -2.3692 ± 0.1054 | -1.5905 ± 0.0034 | -2.1004 ± 0.0256 |
> | + delay          | -2.4018 ± 0.1097 | -1.6023 ± 0.0078 | -2.0291 ± 0.0240 |
> | **GCN-SSM (ours)**| -2.4312 ± 0.0329| -2.8206 ± 0.5654 | -2.2446 ± 0.0027 |
> | + eig(Λ) ≈ 1     | -2.4442 ± 0.0984| -3.5928 ± 0.1026 | -2.2583 ± 0.0085 |
> | + k-hop          | -3.0748 ± 0.0545| -3.6044 ± 0.0291 | -4.2652 ± 0.1776 |
>
> These results highlight that GNN-SSM not only scales well in practice, but also achieves competitive or superior performance across both real-world and synthetic tasks, while maintaining a minimal and interpretable formulation.
>
> > Provide a discussion between the functional differences in the proposed method to avoid vanishing gradient in comparison with the known approaches in literature.
>
> Our approach, unlike other recent methods for mitigating vanishing gradients in GNNs [1–3], gives explicit control over the model's dynamics via the spectrum of the Jacobian. This allows us to directly regulate gradient flow and stability. We will add a brief discussion of this in the revised version, along with a comparison to other more opaque strategies for gradient mitigation, such as those relying on normalization layers or residual scaling, which often lack interpretability.
>
> [1] Alessio Gravina, Davide Bacciu, and Claudio Gallicchio. Anti-Symmetric DGN: A Stable Architecture for Deep Graph Networks. ICLR, 2023.
>
> [2] Alessio Gravina, Moshe Eliasof, Claudio Gallicchio, Davide Bacciu, and Carola-Bibiane Schönlieb. On Oversquashing in Graph Neural Networks through the Lens of Dynamical Systems. AAAI, 2025.
>
> [3] T. Konstantin Rusch, Ben Chamberlain, James Rowbottom, Siddhartha Mishra, and Michael Bronstein. Graph-Coupled Oscillator Networks. ICML, 2023.
>
> [4] Luo, Yuankai, et al. "Enhancing graph transformers with hierarchical distance structural encoding." Advances in Neural Information Processing Systems 37 (2024): 57150-57182.
>
> [5] Gutteridge, Benjamin, et al. "Drew: Dynamically rewired message passing with delay." International Conference on Machine Learning. PMLR, 2023.

---

> > ### Comment · Reviewer_CMgd · 2025-08-02
> >
> > We appreciate the comparison with state of art GCN techniques. It is heartening to see that the proposed methods maintains the superiority of the related techniques. Some of the finer advantages of the proposed method is also clear now. The method does have specific technical advantages.

---

> > > ### Author Response · Authors · 2025-08-05
> > >
> > > We thank the reviewer for their thoughtful review and engagement, and we hope that our clarifications have further strengthened your evaluation toward full acceptance of our work. Further, we remain available to provide any additional clarification during the discussion phase, if needed.

---

> > > > ### Author Response · Authors · 2025-08-08
> > > >
> > > > Given that we are less than 24 hours from the end of the discussion, we would like to thank the reviewer again for their thoughtful review and engagement.
> > > >
> > > > We are pleased that the reviewer appreciated our additional evaluations and recognized the specific technical advantages of our method. In light of this, we hope that the reviewer will consider raising the score in their final decision to reflect their positive view of the current version of our work.

---

> > ### Comment · Reviewer_CMgd · 2025-08-09
> >
> > We are convinced with the scalability results. The method is scalable. The strategy for mitigating vanishing gradients builds on existing approaches. In general, the proposed approach is novel and interesting.

---

### Official Review · Reviewer_v1TX · 2025-07-03

**Clarity:** 3
**Significance:** 2
**Originality:** 2
**Rating:** 4
**Confidence:** 4

**Summary:**

This paper analyzes depth‐related performance drops in GNNs (e.g., over-smoothing, over-squashing) through the lens of vanishing gradients. By vectorizing node features and treating layer-to-layer propagation as a dynamical system, this paper shows that standard graph convolution and attention layers are susceptible to gradient vanishing, whose Jacobians are strongly contractive because of the normalized adjacency operators. This paper introduces GNN-SSM, a state-space model-inspired architecture that explicitly controls the eigenvalues of the state transition matrix. This preserves gradient norms, prevents feature collapse, and, when combined with k-hop aggregation, achieves strong performance on synthetic long-range benchmarks and the LRGB tasks.

**Questions:**

Please see Weaknesses

**Ethical Concerns:**

["NO or VERY MINOR ethics concerns only"]

**Final Justification:**

The authors' rebuttal provides a more detailed explanation of the relation to previous works, which addresses my concerns. Given the new additions in the discussion, I'd like to raise my rating and recommend acceptance.

**Limitations:**

yes

**Quality:**

3

**Strengths And Weaknesses:**

Pros:
- This paper provides a unified perspective that links vanishing gradients, over-smoothing, and over-squashing via spectral bounds of Jacobians
- To validate the theoretical analysis, this paper designs a simple remedy to the gradient-vanishing problem by making the eigenvalues of the state transition matrix close to 1, inspired by state-space models, and further combines k-hop aggregation to address over-squashing. The model's good empirical performance validates the theory

Cons:

My major concern with this paper is that its contribution/novelty compared to existing works is unclear.

Theory side, the theoretical analysis of this paper has a lot of overlaps with previous studies. E.g., [1] analyzed over-smoothing from the dynamical system's point of view and reached similar conclusions about convergence to fixed points. [2] explored the connection between gradient-vanishing and over-squashing, similarly to this paper.

Model design side, this paper proposes to connect SSMs and GNNs. This idea has already been explored in the graph SSM literature. Especially, the GNN-SSM architecture proposed by this paper is very similar to GRED [3]. GRED has already used LRU to bound the eigenvalues of the state transition matrix for stable signal propagation, and combined k hops to address over-squashing (similar to kGNN-SSM). However, the relation to GRED seems to be downplayed in this paper, lacking proper comparison/discussion

[1] Graph Neural Networks Exponentially Lose Expressive Power for Node Classification. ICLR 2020.
[2] On Over-Squashing in Message Passing Neural Networks: The Impact of Width, Depth, and Topology. ICML 2023.
[3] Recurrent Distance Filtering for Graph Representation Learning. ICML 2024

---

> ### Author Rebuttal · Authors · 2025-07-30
>
> We would like to thank you very much for your thorough review. We also thank you for the excellent points you bring up, we would like to address them below.
>
> >**Weakness 1** Theory side, the theoretical analysis of this paper has a lot of overlaps with previous studies. E.g., [1] analyzed over-smoothing from the dynamical system's point of view and reached similar conclusions about convergence to fixed points. [2] explored the connection between gradient-vanishing and over-squashing, similarly to this paper.
>
> Thank you for raising these great points. While we agree with the reviewer that there are similarities with prior work in the oversmoothing literature, we emphasize several key differences between our contributions and existing research:
>
> (i) We argue that the primary issue limiting depth in GNNs is not oversmoothing per se, but rather trainability, specifically, the problem of vanishing (and sometimes exploding) gradients. In fact, the well-known feature collapse phenomenon arises as a consequence of vanishing gradients. In our view, GNNs fail at large depths due to extreme trainability issues, not a loss of expressivity. To the best of our knowledge, this connection has not been explicitly discussed in prior work.
>
> (ii) The analysis in [1] is limited to GCNs and GATs. Our theoretical study also begins with these models to illustrate the severity of gradient vanishing in these models via Jacobian analysis. However, our use of layer-wise Jacobians to assess both network trainability and the degree of smoothing is novel, and can be applied to gain insight into any GNN layer that might be difficult to study mathematically. As an example of this, and in light of your suggestion and that of Reviewer ZDHY, we analyzed the Jacobian spectrum of a randomly initialized GIN layer and observed significant instability, with spectral norms reaching approximately 150 (much larger than the region of stability around 1), which would lead to exploding gradients when stacked together. This leads to divergent Dirichlet energy with more layers, which explains behaviors recently discussed in [3]. It is this instability, rather than the appearance of smoothing, that hampers GIN’s trainability.
>
> We will include this analysis in the paper along with other common GNN architectures (e.g. Gated-GCN) to illustrate that our point is not restricted to GCNs.
>
> Regarding our theoretical treatment of over-squashing, we highlight key distinctions from [2]:
>
> (i) We explicitly connect oversmoothing and over-squashing via vanishing gradients. Building on our earlier results, we argue that even if two nodes are able to communicate perfectly at a given layer  $l$, this communication becomes ineffective if the messages have already collapsed to an uninformative fixed point. We demonstrate the importance of this effect in Figure 4. This connection, to our knowledge, was left open in [2], which states:
>
> > “On a different note, this work has not touched on the important problem of over-smoothing; we believe that the theoretical connections we have derived, based on the relation between over-squashing, commute time, and Cheeger constant, suggest a much deeper interplay between these two phenomena.”
>
> (ii) While [2] primarily focuses on rewiring to improve connectivity, we argue that it is equally important to prevent vanishing gradients through model design. We note that existing architectures that perform well in long-range tasks (e.g., Drew or GRED) implicitly address both aspects, although they do not explicitly state the importance of this combination or the reasons behind it. We test this through the kGCN-SSM framework, which was designed to be as minimal and controllable as possible, while still allowing us to isolate and showcase the role of effective rewiring and stable gradients. Our ablation studies confirm the necessity of these individual components.
>
> This perspective contrasts with [2], which explicitly cautions against mitigating vanishing gradients via the model term:
>
> > “However, this is not an optimal solution since increasing the contribution of the model (i.e., the term $c_{\sigma} w p$ may lead to overfitting and poorer generalization (Bartlett et al., 2017). Taking larger values of $c_{\sigma}, w, p$ affects the model globally and does not target the sensitivity of specific node pairs induced by the topology...”
>
> Moreover, [2] hints toward the need to explore ideas from RNNs in this context:
>
> > “We believe that the relation between over-squashing and vanishing gradients deserves further analysis. In particular, it seems that there is a phase transition that MPNNs undergo—from over-squashing of information between distant nodes, to vanishing of gradients at the level of the loss. In fact, this connection suggests that traditional methods used in RNNs to mitigate vanishing gradients may also be beneficial for over-squashing.”
>
> Our work directly addresses and bridges the above gaps by offering theoretical insights and a minimal framework with which to prevent the effects of gradient vanishing.
>
> >**Weakness 2:** Model design side, this paper proposes to connect SSMs and GNNs. This idea has already been explored in the graph SSM literature. Especially, the GNN-SSM architecture proposed by this paper is very similar to GRED [3]. GRED has already used LRU to bound the eigenvalues of the state transition matrix for stable signal propagation, and combined k hops to address over-squashing (similar to kGNN-SSM). However, the relation to GRED seems to be downplayed in this paper, lacking proper comparison/discussion
>
> We would like to clarify that the primary goal of our model design is not to achieve state-of-the-art performance (as is the case with GRED) but rather to construct a minimal and controllable framework that allows us to isolate and study specific phenomena we are interested in. In particular, GNN-SSM was designed to investigate the influence of vanishing gradients, while kGNN-SSM extends this to jointly analyze the impact of vanishing gradients and rewiring. For this reason, the majority of our experiments take the form of targeted ablations rather than performance benchmarks.
>
> The k-hop rewiring scheme was deliberately chosen to be similar to that of DRew [5], with the distinction that in our setting, a node at layer $l$ only aggregates information from nodes exactly $l$-hops away. This choice allowed us to construct a strong baseline that supported an independent evaluation of gradient stability and its effect on over-squashing. As the reviewer notes, this also bears resemblance to GRED; however, we emphasize that such multi-hop aggregation schemes predate GRED or Drew, for instance, ChebNet [4], which is one of the earliest GNN designs, uses a very similar aggregation scheme.
>
> Having said this, we also note several important differences:
>
> (i) **Aggregation scope**: GRED aggregates information from several neighborhoods at each layer. In contrast, our approach aggregates from a **single-hop neighborhood** per layer.
>
> (ii) **Aggregation direction**: GRED’s SSM update operates inward, beginning with distant nodes and moving toward the root node. In our model, each additional layer aggregates from progressively more distant neighbors, moving outward. This pattern is more consistent with the standard MPNN framework.
>
> (iii) **Parameterization of the state matrix**: In our model, the state matrix is fixed and not learned, eliminating the need for additional constraints to guarantee stable learning dynamics.
>
> (iv) **Weight sharing**. We do not perform weight sharing when performing k-hop aggregation, as in GRED.
>
> We stress that our intention was not to downplay GRED, but rather to conduct a fair evaluation. We found our model to be more related to DRew (due to their similar rewiring and information processing schemes), which made the comparison much more amenable to ablation. In particular, in DRew, we can directly assess the impact of the model term by removing the delay term, while doing so in GRED was slightly more complex. For example, GRED is wrapped around a “Transformer-type block” structure, which we thought would make it hard to disentangle what was driving performance exactly, and would potentially limit the validity of a controlled ablation.
>
> Nonetheless, in response to the reviewer’s comment, we now include the following results on the Peptides dataset:
>
> | Dataset           | Model              | Performance (Mean ± Std) |
> |------------------|--------------------|---------------------------|
> | Peptides-func     | kGCN-SSM (small \(d\)) | 69.02 ± 0.218             |
> |                  | kGCN-SSM (large \(d\)) | 72.12 ± 0.268             |
> |                  | GRED                | 70.85 ± 0.27              |
> | Peptides-struct   | kGCN-SSM (small \(d\)) | 28.98 ± 0.324             |
> |                  | kGCN-SSM (large \(d\)) | 27.01 ± 0.071             |
> |                  | GRED                | 25.03 ± 0.19              |
>
> We will also include a more extensive discussion of these comparisons in the revised version, as well as more detailed ablations on both the Peptides and GPP tasks.
>
> In light of the clarified contributions above, we kindly ask the reviewer to consider raising their score or let us know whether there is further improvement that will lead to a more favourable evaluation of our work.
>
> [1] Graph Neural Networks Exponentially Lose Expressive Power for Node Classification. ICLR 2020.
>
> [2] On Over-Squashing in Message Passing Neural Networks: The Impact of Width, Depth, and Topology. ICML 2023.
>
> [3] Arnaiz-Rodriguez, A., et al.. "Oversmoothing," Oversquashing", Heterophily, Long-Range, and more: Demystifying Common Beliefs in Graph Machine Learning. arXiv preprint arXiv:2505.15547 (2025).
>
> [4] Defferrard, M., et. al. "Convolutional neural networks on graphs with fast localized spectral filtering." NeurIPS 2016.
>
> [5] Gutteridge, B., et al. "Drew: Dynamically rewired message passing with delay." ICML 2023.

---

> > ### Comment · Reviewer_v1TX · 2025-08-08
> >
> > I appreciate the authors' detailed response. Further explanations make the relation to previous works clearer. I'm convinced to increase my rating and recommend acceptance. I strongly suggest that the authors reflect the new additions in the updated version to avoid readers having similar confusion.

---

### Decision · Program_Chairs · 2025-09-17

**Decision:**

Accept (poster)

**Comment:**

Based on ideas from linear control theory, the paper presents an unified view of the problem of over-smoothing/over-squashings in GNNs through the lens of vanishing gradients. It interprets GNNs as recurrent models and empirically demonstrates that a simple state-space formulation  effectively alleviates these issues without introducing additional trainable parameters. Moreover, the paper shows theoretically and empirically that traditional GNNs are by design prone to extreme gradient vanishing, feature collapse is directly related to the mechanism causing vanishing gradients, and long-range modeling is most easily achieved by a combination of graph rewiring and vanishing gradient mitigation.

In the rebuttal the authors convincingly replied to the concerns of the reviewers. The provided a more detailed explanation of the relation and a comparison to previous related works, experimental results analysing the scalability of the proposed architecture, new experiments on the heterophilic datasets from [Platonov2023] and on ogbn-arxiv, demonstrating that their proposed model has good results on node-level tasks. Moreover, the authors have further analyzed the role of the input matrix B. Based on these additional insights, which should be included in the revised version of the paper, all reviewers voted for acceptance.